# Alterations in homologous recombination repair genes in prostate cancer brain metastases

Antonio Rodriguez-Calero[1,2,23], John Gallon [3,23], Dilara Akhoundova[1,4], Sina Maletti[1], Alison Ferguson [1,5], Joanna Cyrta[6], Ursula Amstutz[7], Andrea Garofoli[8], Viola Paradiso[8], Scott A. Tomlins[9], Ekkehard Hewer [2,22], Vera Genitsch[2], Achim Fleischmann[2,10], Erik Vassella[2], Elisabeth J. Rushing [11], Rainer Grobholz [12], Ingeborg Fischer[12], Wolfram Jochum[13], Gieri Cathomas[14], Adeboye O. Osunkoya [15], Lukas Bubendorf [8], Holger Moch [16], George Thalmann[17], Charlotte K. Y. Ng [1], Silke Gillessen[18,19,20,24], Salvatore Piscuoglio [3,8,24✉] & Mark A. Rubin [1,21,24✉]

Improved survival rates for prostate cancer through more effective therapies have also led to an increase in the diagnosis of metastases to infrequent locations such as the brain. Here we investigate the repertoire of somatic genetic alterations present in brain metastases from 51 patients with prostate cancer brain metastases (PCBM). We highlight the clonal evolution occurring in PCBM and demonstrate an increased mutational burden, concomitant with an enrichment of the homologous recombination deficiency mutational signature in PCBM compared to non-brain metastases. Focusing on known pathogenic alterations within homologous recombination repair genes, we find 10 patients (19.6%) fulfilling the inclusion criteria used in the PROfound clinical trial, which assessed the efficacy of PARP inhibitors (PARPi) in homologous recombination deficient prostate cancer. Eight (15.7%) patients show biallelic loss of one of the 15 genes included in the trial, while 5 patients (9.8%) harbor pathogenic alterations in *BRCA1/2* specifically. Uncovering these molecular features of PCBM may have therapeutic implications, suggesting the need of clinical trial enrollment of PCBM patients when evaluating potential benefit from PARPi.

[1] Department for BioMedical Research, University of Bern, Bern, Switzerland. [2] Institute of Pathology, University of Bern, Bern, Switzerland. [3] Visceral Surgery and Precision Medicine Research Laboratory, Department of Biomedicine, University of Basel, Basel, Switzerland. [4] Department of Medical Oncology and Hematology, University Hospital Zurich, Zurich, Switzerland. [5] Department of Oncology, Ludwig Cancer Centre, University of Lausanne, Lausanne, Switzerland. [6] Department of Pathology, Institut Curie, University Paris Sciences et Lettres, Paris, France. [7] Department of Clinical Chemistry, Inselspital, Bern University Hospital, University of Bern, Bern, Switzerland. [8] Institute of Medical Genetics and Pathology, University Hospital Basel, University of Basel, Basel, Switzerland. [9] Departments of Pathology and Urology, University of Michigan Medical School, Ann Arbor, MI, USA. [10] Institute of Pathology, Cantonal Hospital Thurgau, Münsterlingen, Switzerland. [11] Institute of Neuropathology, University Hospital Zurich, Zurich, Switzerland. [12] Institute of Pathology, Cantonal Hospital Aarau, Aarau, Switzerland. [13] Institute of Pathology, Cantonal Hospital St. Gallen, St. Gallen, Switzerland. [14] Institute of Pathology, Cantonal Hospital Baselland, Liestal, Switzerland. [15] Departments of Pathology and Laboratory Medicine, and Urology, Emory University School of Medicine, Atlanta, USA. [16] Department of Pathology and Molecular Pathology, University Hospital Zurich, Zurich, Switzerland. [17] Department of Urology, Inselspital, Bern University Hospital, Bern, Switzerland. [18] Faculty of Biomedical Sciences, USI, Lugano, Switzerland. [19] Department of Oncology, Cantonal Hospital St. Gallen, St. Gallen, Switzerland. [20] Division of Cancer Sciences, University of Manchester, Manchester, United Kingdom. [21] Bern Center for Precision Medicine, Inselspital, Bern University Hospital, University of Bern, Bern, Switzerland. [22] Present address: Institute of Pathology, Lausanne University Hospital and University of Lausanne, Lausanne, Switzerland. [23] These authors contributed equally: Antonio Rodriguez-Calero, John Gallon. [24] These authors jointly supervised this work: Silke Gillessen, Salvatore Piscuoglio, and Mark A. Rubin. ✉email: s.piscuoglio@unibas.ch; mark.rubin@dbmr.unibe.ch

L
ethal prostate cancer (PCa) commonly metastasizes to bone, lymph nodes and, less frequently, to visceral organs[1]. In the largest survey of over 550 autopsy cases of metastatic PCa, prostate cancer brain metastases (PCBM) were identified in only 1.5% of cases[1]. In contrast, brain metastases in other cancers are more common (e.g., 16.3% in lung, 9.8% in renal cell carcinoma, 7.4% in melanoma, or 5% in breast cancer[2,3]). Large-scale genomics studies of metastatic castration-resistant PCa (mCRPC) have revealed the enrichment for TP53 and AR alterations in PCa metastases compared to primary disease[4] and the association between RB1 mutations and mCRPC outcome[5,6]. The recent improvement in systemic therapy for PCa has led to significantly increased patient survival, with average survival extended to about 40 months compared to 15 months with earlier therapies (reviewed in[2]). However, with prolonged survival, oncologists have noted an increased occurrence of PCBM[7]. We posited that PCBM may require distinct genetic changes that distinguish these tumors from more common PCa metastases. As little is known about the genomic landscape of these rare metastases, we conducted a comprehensive multi-regional genomic analysis of a PCBM cohort of 51 patients, with non-synchronous matched primary samples available for 20 patients.

In this study, we comprehensively assess the genomic features which define PCBM.

In order to provide insights into mutational processes specifically associated with metastasis to the brain we compare primary samples from this cohort to the large scale primary-PCa cohort from The Cancer Genome Atlas (TCGA)[8] and metastatic PCBM samples to mCPRC from other anatomic sites from the Stand Up to Cancer/Prostate Cancer Foundation castration-resistant prostate cancer cohort (CRPC500)[4,5,9]. For 20 matched cases with both primary and metastatic samples, we identify putative driver genetic events resulting from clonal evolution that could drive metastasis and examine the clonal evolution occurring in cases with multiple intratumoral regions within the primary tumor and PCBM. We further compare pathways affected by genetic alterations in patients harboring parenchymal brain metastases with those presenting dural metastases in order better to understand the processes which may determine the specific site of metastasis.

## Results

**Demographic and clinical data of the PCBM cohort**. The cohort of 51 patients analyzed here (Supplementary Table 1, Supplementary Fig. 1), represents a substantial increase in the number of PCBM samples over existing studies. (Supplementary Table 2). The average age at the time of PCBM diagnosis among the 51 patients was 71 years. 56% (29/52) harbored multiple and 41% (21/51) singular CNS metastasis. Metastases in brain parenchyma were present in 41% (21/51), dural metastases in 35% (18/51) while in 24% (12/51) of the patients the primarily metastatic location was either unknown or unclear by involvement of multiple anatomical structures. Additionally, 88% (45/51) of the patients presented non-brain metastases with bone involvement in 91% (41/45) of these. Androgen deprivation therapy or orchiectomy were conducted in 82% (42/51) of the cases. From those, 26% (11/42) underwent further therapy with next-generation ARSi (androgen receptor signaling inhibitors), namely abiraterone and/or enzalutamide (Supplementary Table 1, Supplementary Fig. 1).

**Histomorphology of primary tumors and brain metastases**. We first reviewed the histomorphology across the PCBM cohort. Pure acinar adenocarcinoma histology was identified in 48/51 (94%) PCBM and in 20/20 (100%) primary PCa. In 2/20 (10%) of the

primary PCa samples, we also identified focal neuroendocrine (NE) differentiation by IHC. The remaining 3/51 (6%) PCBM showed either areas of small cell NE carcinoma admixed with acinar adenocarcinoma (observed in patient P27), or features intermediate between NE carcinoma and acinar adenocarcinoma (patients P1 and P33)[10]. This distribution of morphologic phenotypes is similar to a recent study by Abida et al.[5] where 89% of CRPC cases were classified as adenocarcinoma and 11% showed NE features. The majority of primary PCa contained high-grade areas consisting of ISUP-Grade Group 5 (15/20; 75%). The remaining cases were Grade Group 4 (2/20; 10%) or Grade Group 3 (1/20; 5%), while two tumors were not gradable. When enough tissue was available, we performed IHC analysis for protein expression of frequently altered genes in PCa (ERG, p53, and PTEN) to identify tumor heterogeneity. Based on the morphology and immunohistochemical profile, one or more intratumoral regions of interest (ROIs) were defined and sampled for subsequent genomic investigation (WES/targeted RNA), totaling 168 samples (105 from PCBM, 63 from primary PCa) from 51 patients (Fig. 1a, Supplementary Fig. 1, and Supplementary Data 1).

**The genetic landscape of PCBM**. Across the 168 samples, we detected an average of 4.43 coding alterations/Mb, including SNVs and indels (range 0.28–50.45). We detected an average of 2.98 SNVs/Mb (range 0.22–38.21), 0.49 deletions/Mb (range 0–10.22) and 0.15 insertions/Mb (range 0–2.02). Significantly more SNVs, insertions and deletions, were detected in PCBM compared to the matched primaries (SNVs $q = 6.75 \times 10^{-6}$, insertions $q = 3.06 \times 10^{-5}$, deletions $q = 9.30 \times 10^{-3}$, Wilcoxon test) (Fig. 1b, Supplementary Fig. 2a, b).

We found significantly higher levels of SNVs, insertions and deletions in the primary samples of PCBM cohort compared to TCGA (SNVs $q = 1.21 \times 10^{-22}$, insertions $q = 1.58 \times 10^{-3}$ and deletions $q = 5.67 \times 10^{-10}$, Wilcoxon test). As the primary PCa from PCBM cohort were mainly high grade (ISUP-Grade Group, GG > 3), we performed the same comparison using the high grade (GG > 3) TCGA samples and found similar enrichment for all alteration types in the PCa of PCBM cohort (SNV $q = 8.78 \times 10^{-18}$, insertions $q = 4.03 \times 10^{-2}$ and deletions $2.19 \times 10^{-7}$, Wilcoxon test) (Supplementary Fig. 2c). Comparing coding alterations in PCBM against the CRPC500 showed a significant enrichment for somatic alterations compared to non-brain metastases (SNVs $q = 2.38 \times 10^{-19}$, insertions $q = 2.82 \times 10^{-15}$ and deletions $q = 8.44 \times 10^{-14}$, Wilcoxon test) (Supplementary Fig. 2a, b). These differences persisted after two analyses examining the effect of different sequencing depths across cohorts, first by applying more stringent alternate allele read thresholds (Supplementary Fig. 3a), and secondly by simulating the effects of downsampling our sequencing data (Supplementary Fig. 3b). We also observed the same enrichments when making these comparisons using only the cases for which we had matched normal tissue (Supplementary Fig. 3c). This confirmed that the differences in detected mutations are not as a result of different sequencing depths between cohorts, or the inclusion of samples without a matched normal.

The average total coding mutation rate across the whole cohort was below 15 mutations/Mb for all but four patients (patients P39, P58 P48, and P55) (Fig. 1b). Patients P39, P58, and P48 had high representation of the SBS44 (defective DNA mismatch repair) Single Base Substitution (SBS) signature. Of the three samples with high representation of SBS44, samples from patients P39 and P48 harbored missense and frameshift mutations, respectively, in MSH2 and both showed high microsatellite instability (MSI-H) status using a clinical molecular diagnostic

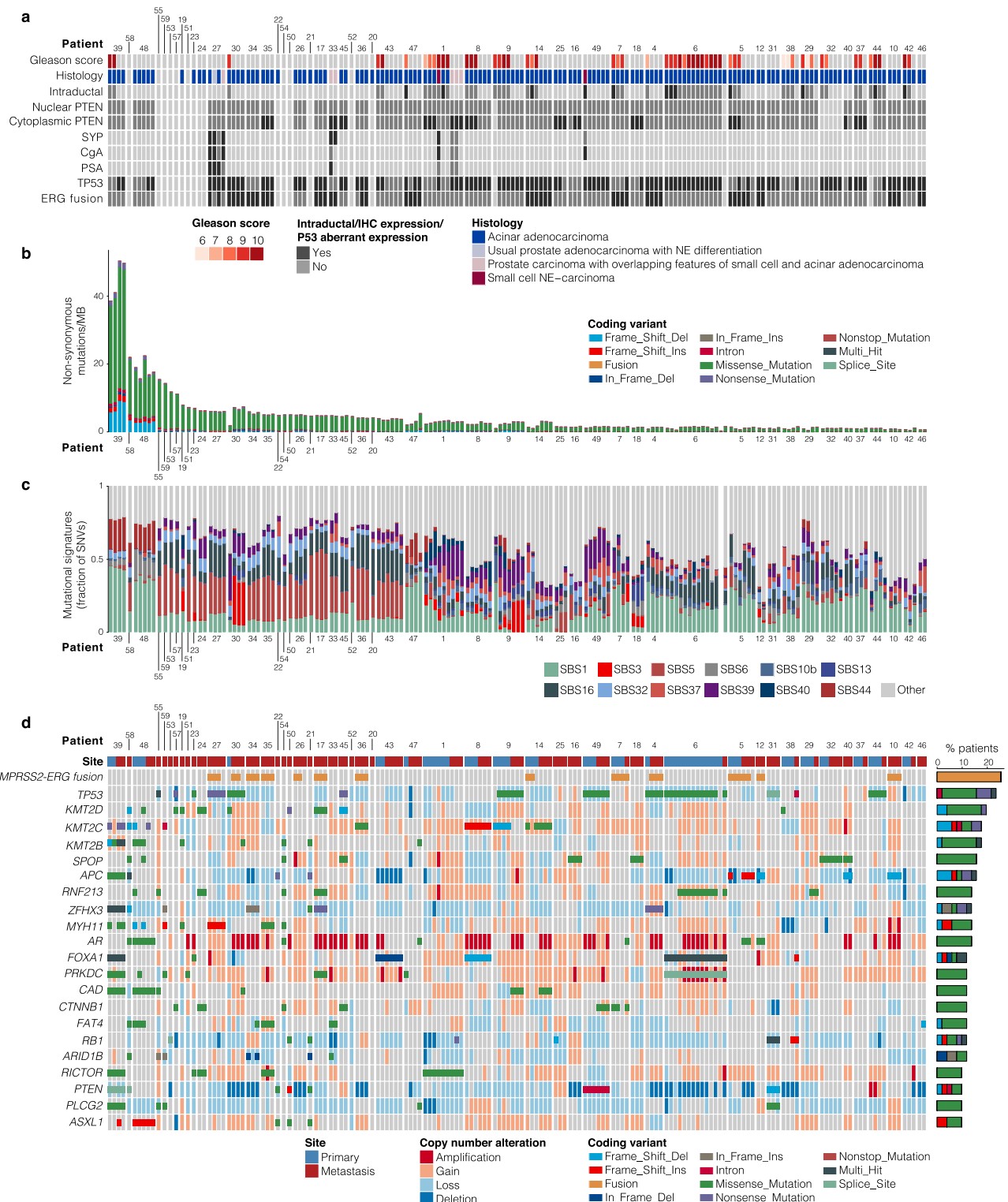

**Fig. 1 Summary of histologic and genetic alterations in prostate cancer brain metastases. a** Histology and immunohistochemistry results, **b** Summary of non-synonymous mutations/Mb. **c** Relative contribution of mutational signatures from COSMIC. **d** Summary of genes showing recurrent mutation or fusion (altered in >5 patients) from WES and targeted RNA-seq data. CNA are shown in large blocks, coding alterations in small blocks. Alteration types are colored according to the legend. Samples are grouped by patient and site (primary or metastasis). Source data are provided in Supplementary Tables 4 and 5.

assay[11] (Supplementary Fig. 4). Samples from patient P58 had a frameshift variant in *POLD1* and a missense variant in *RFC4*, and those from patient P55 harbored missense variants in *POL3RC* and *RNMT*, but no alteration affecting an MMR or HRR gene. However, SBS10b was detected in patient P55, associated with large numbers of mutations in samples defined as 'hypermutators'[12]. Signatures associated with ageing (SBS1 and SBS5) were the most frequently observed across the cohort, detected in 19 and 22 (37 and 43%) patients, respectively, reflecting the advanced age of the patients[13] (71 years old median age at PCBM diagnosis) (Fig. 1c, Supplementary Fig. 5). Following these, SBS16 (strand bias of DNA damage and nucleotide excision repair) was observed in 19 (37%) patients.

In the metastases from the PCBM cohort, the most frequently mutated genes were *TP53* (13 patients, 25%), *APC* (eight patients, 16%), and *SPOP* (eight patients, 16%). As expected, given that 43 patients (84%) had received androgen deprivation therapy and/or abiraterone/enzalutamide, we also detected frequent *AR* alterations, with seven patients (14%) showing coding alterations and 32 patients (63%) showing *AR* amplifications. We detected *PTEN* loss or deletion in 33 patients (64.71%) (Fig. 1d, Supplementary Data 2, Supplementary Data 3). Targeted RNA-sequencing identified the *TMPRSS2-ERG* fusion in 13 patients (25%), which correlated with ERG overexpression by IHC (Pearson's correlation 0.70, $P < 2.2 \times 10^{-16}$) (Fig. 1a, d, Supplementary Data 2).

**Homologous recombination deficiency is prevalent in PCBM.** Analysis of mutational signatures showed a high representation of the HRD mutational signature SBS3 in four patients (SBS3 > 10% mutations, P7, P9, P19, and P30) (Fig. 1c). The HRD mutational signature (SBS3) is enriched in brain metastases from colorectal cancer[14], has been detected in primary and metastatic PCa, independent of *BRCA1/2* alterations[15] and has been suggested to contribute to the accurate prediction of response to PARP-inhibitor treatment in combination with the presence of alterations affecting HRR genes[16]. Therefore, we investigated the prevalence of the HRD mutational signature within the cohort, as well as in TCGA and CRPC500 cohorts, using the computational tool Signature Multivariate Analysis (SigMA) for the highly sensitive detection of the HRD signature[17]. We found significantly greater representation of the HRD signature in PCBM compared to CRPC500 ($q = 0.041$, Wilcoxon test), and in primary samples of the PCBM cohort compared to TCGA (Fig. 2a) ($q = 0.0003$, Wilcoxon test), even when comparing only to the subset of high-grade TCGA samples (GG > 3) ($q = 0.0034$, Wilcoxon test), (Supplementary Fig. 6). Given this enrichment in the PCBM cohort, we focused on alterations affecting at least one of the 15 homologous recombination repair (HRR) genes included in the PROfound clinical trial[18] (*BRCA1, BRCA2, ATM, BRIP1, BARD1, CDK12, CHEK1, CHEK2, FANCL, PALB2, PPP2R2A, RAD51B, RAD51C, RAD51D,* and *RAD54L)* (referred to hereafter as PROfound genes). In the metastatic samples from the PCBM cohort, 51 patients (100%) harbored some genetic alteration, either mutation or somatic copy number alterations (loss/deletion), in at least one of those genes.

Moreover, HRR alteration, defined by the detection of the HRD signature by SigMA and/or copy number loss /deletion or mutation affecting PROfound genes[18], was enriched in PCBM compared to CRPC500, with 100% and 87.26% of patients affected, respectively ($P = 0.0025$, Fisher's exact test) (Supplementary Fig. 7a). A trend towards higher levels of HRR alteration was also found in primaries of the PCBM-cohort compared to TCGA high GG samples ($P = 0.052$, Fisher's exact test) (Supplementary Fig. 7b). However, many of these alterations are not known to be clinically relevant.

Therefore, we focused on known pathogenic alterations affecting PROfound genes. We found ten patients (19.6%) fulfilling the inclusion criteria used in the PROfound clinical trial of PARP inhibitors (PARPi) in prostate cancer, with eight of these (15.7%) showing biallelic loss of one of the 15 genes included in the trial, while five patients (9.8%) harbored well documented pathogenic alterations of *BRCA1/2* specifically (Fig. 2b, Supplementary Table 3, Supplementary Data 2 and Supplementary Data 3).

Taken together, these data highlight the prevalence of HRD in PCBM with implications for patient stratification and treatment. Most notably, the subset of ten patients meeting the inclusion criteria for the PROfound trial, represent a group of PCBM patients who could have been enrolled in a study evaluating benefit from PARPi.

**Clonal evolution in PCBM.** The 20 cases for which we had both primary and metastatic samples provide a unique opportunity to study the clonal evolution occurring in primary prostate cancers which metastasized to the brain, and in the metastases themselves. Most other studies of CRPC have only rarely had more than a few non-synchronous, matched primary and metastatic samples to explore. Furthermore, given the inclusion of patients with both dural and parenchymal metastases, these samples allow us to interrogate the nature of the clonal evolution specific to each of these locations. Therefore, using the cancer cell fraction (CCF) estimates for synonymous and non-synonymous SNVs from ABSOLUTE[19] and PhylogicNDT[20], we examined clonal evolution within and between samples from the primary and metastatic sites. We calculated clusters of mutations in each patient, and the CCF of these clusters in each patient sample from all 20 cases. Furthermore, we were able to generate clonal evolution trees from 14 of these patients (Fig. 3a–d, Supplementary Fig. 8). The CCFs across primary and metastatic samples for the remaining six patients are shown in Supplementary Fig. 9.

In 18/20 cases we observed the expansion of clones from subclonal in the primary samples, to a clonal level (CCF > 0.9) in the metastatic samples (Supplementary Fig. 8, Supplementary Fig. 9). We highlight the clonal evolution occurring in two patients with dural and two with parenchymal metastases in detail (patients P4 and P14, and P5 and P8, respectively) (Fig. 3a–d). In samples from patient P4, which had truncal mutations in *ZFHX3, TP53,* and *POU2F,* cluster 2 (light blue) mutations became clonal in all three metastatic samples, and cluster 4 (expanded to a clonal level in sample M2 only (Fig. 3a)). In patient P14, samples from which did not show a truncal mutation in a driver gene, cluster 4 (dark green) showed an increasing cellular fraction across primary samples (sample P1 0.19, P2 0.36, P3 0.76) and expanded to become clonal in metastatic samples M2 and M3 (CCF 0.95), and near clonal in M1 (CCF 0.86) (Fig. 3b). From these clones, cluster 5 (light purple), which emerged at a subclonal level in sample P3 (CCF 0.54), expanded to become clonal in sample M2, and to a near clonal level in samples M1 and M3 (CCF 0.83 and 0.88 respectively). Cluster 7 (light green), similarly expanded to become clonal in M2 and M3 (CCF 0.95, 0.97 respectively) and near clonal in M1 (CCF 0.86). Interestingly, in this patient, clones with a cluster of mutations (cluster 6, dark red) present at a clonal level in samples P2 and P3 were not present at a high CCF in the metastatic samples (CCFs M1 0.15, M2 0.05, and M3 0.05), but evolved a further clonal cluster (cluster 8, pink) of mutations in primary samples P1 and P3.

The representative samples from patients with parenchymal metastases (patients P5 and P8) demonstrated similar clonal evolutionary processes as seen in the dural metastases. Patient

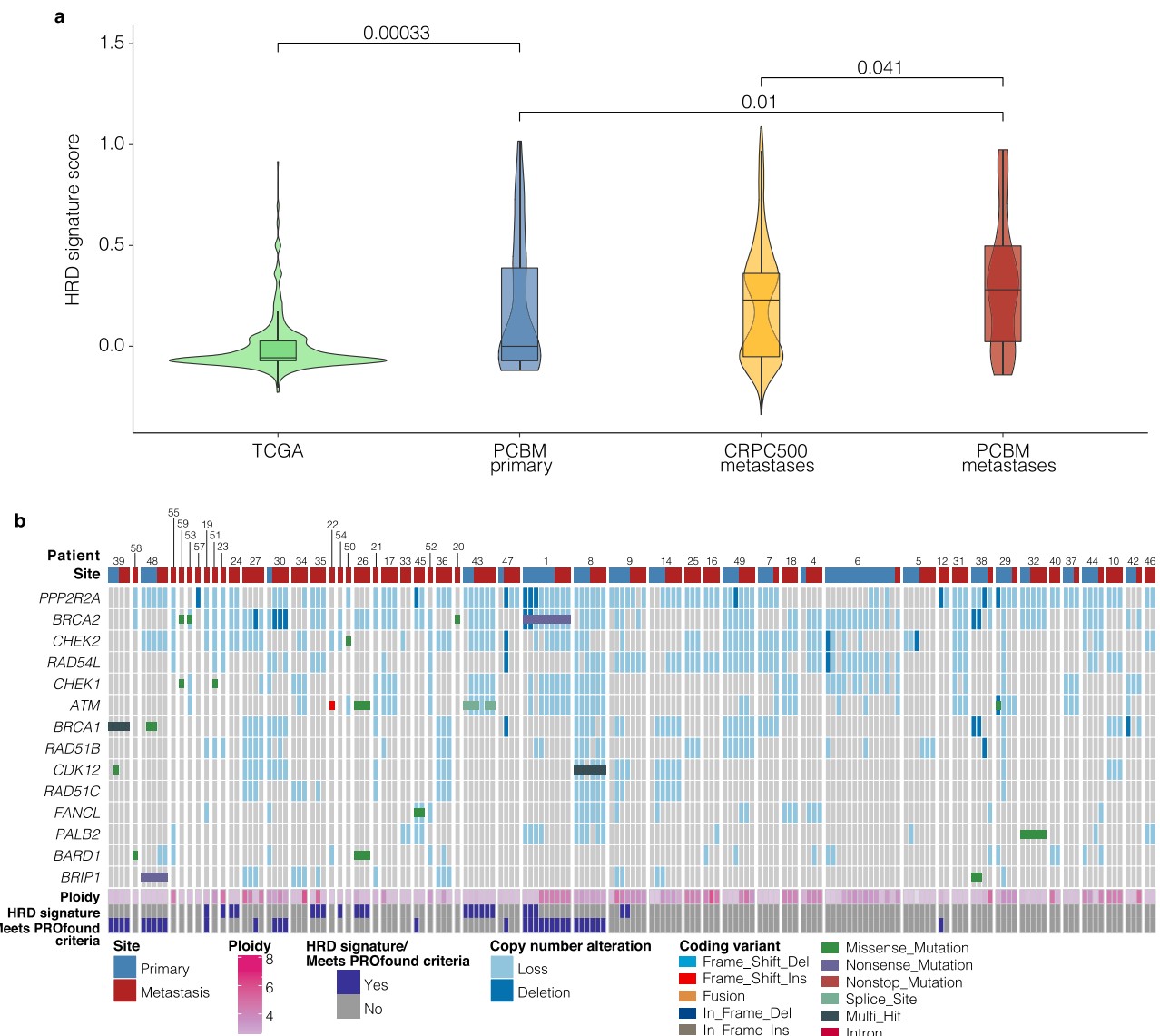

**Fig. 2 HRR genes alterations (PROfound genes) are highly represented and the HRD signature is enriched in PCBM. a** Comparison of HRD signature scores, estimated by SigMA for TCGA (n = 495), PCBM primary (n = 63), CRPC500 non-brain metastatic (n = 411), and PCBM metastatic (n = 105) samples (Wilcoxon test, two sided). Horizontal lines in boxplots show median, hinges show interquartile range, whiskers show 1.5 x interquartile range, points beyond 1.5 IQR past hinge are shown. **b** Summary of mutations and copy number alterations affecting HR genes included in the PROfound clinical trial. Alterations (coding and copy number) are colored according to the legend. Ploidy annotation from FACETS and samples passing the HRD Signature 'strict' cutoff from SigMA are indicated. Source data are provided as a Source Data file.

P5 showed no truncal mutation in a common driver gene (Fig. 3c), while samples from patient P8 harbored truncal mutations in *CDK12*, *EPHA2*, and *KMT2C* (Fig. 3d). In patient P5, cluster 2 was present at a clonal level (CCF 0.96) in primary sample P1, and at lower CCFs (0.27 and 0.35) in primary samples P2 and P3. Clones with this cluster of mutations were, however, clonal or near-clonal in all three metastatic samples (CCF M1 0.88, M2 0.92, M3, 0.97), suggesting the metastases may have arisen from clones present at the region from which the P1 sample was taken. Interestingly, we observed a branching event stemming from these clones, with clones harboring cluster 3 and 6 mutations becoming clonal in metastatic samples M1 and M2 (CCF M1 1.00, M2 0.98). In metastatic sample M3, however, these mutations were only present at a CCF of 0.01, whereas clones containing cluster 3 and cluster 7 mutations, which included an *ASXL2* variant, expanded to a clonal level (CCF 0.99). Similarly, in patient P8, clones with cluster 3 and 4 mutations

became clonal in all three metastatic samples (CCF cluster 3 M1 1.00, M2 1.00, M3 1.00, cluster 4 M1 1.00, M2 1.00, M3 1.00) and, in which further mutations were acquired (cluster 6) which expanded to a clonal level in sample M1 only (CCF M1 1.00, M2 0.50, M3 0.02).

These cases, along with the other ten for which we were able to obtain multiple intrametastatic samples, demonstrate the heterogeneity present in the metastases. For example, in patient P14 cluster 10 was present at CCFs of 0.27, 0.02, and 0.79 in samples M1, M2, and M3, respectively (Fig. 3b), while the metastatic samples from patient P9 showed the emergence of a clonal cluster of mutations in samples M1 and M3 (cluster 6), which was not present in sample M2, where instead cells with cluster 8 mutations were present at a clonal level (Supplementary Fig. 9b).

In the literature, the few PCBM that have been studied are mostly tumors associated with the dura which have not invaded

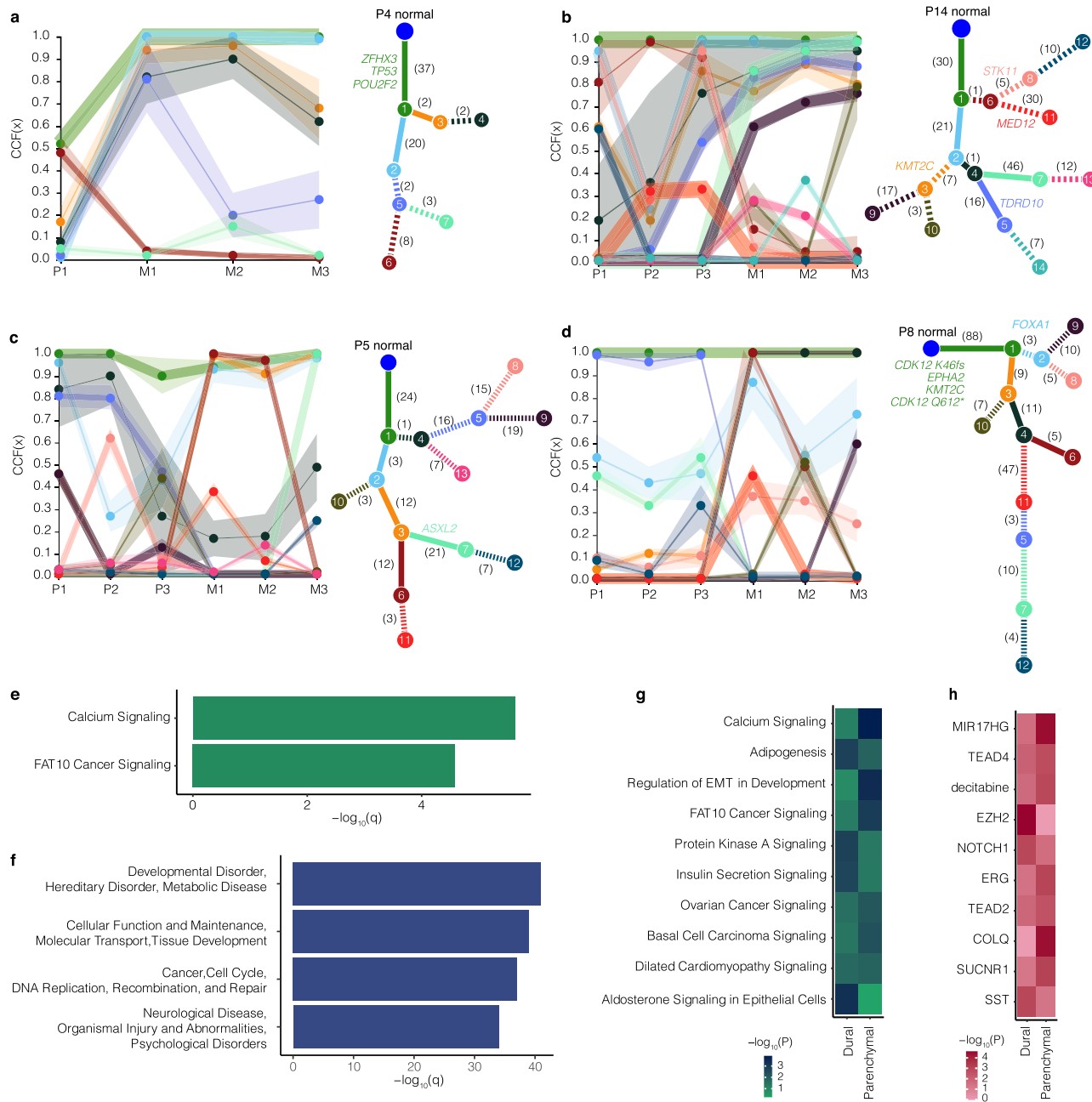

**Fig. 3 Clonal evolution in PCBM Clonal evolution in four patients from the PCBM cohort.** Patients P4 (**a**) and P14 (**b**) had dural metastases, and patients P5 (**c**) and P8 (**d**) had parenchymal metastases. Trace plots show cancer cell fraction (CCF) for each mutational cluster in each patient sample (P = primary, M = metastasis). Ribbons show 95% confidence interval, center of bands show mean cluster CCF estimate. Phylogenetic trees show best solution for evolutionary relationship between clones with different clusters of mutations where each node (numbered) is a cluster of mutations. Numbers on each branch show the number of mutations distinguishing a clone from the previous (all genes). Potential driver genes mutated in the distinction between a clone and the previous are indicated in colors corresponding to the branch. Solid branches show clusters of mutations which become clonal in metastatic samples. **e** Enrichment for canonical pathways associated with genes in mutational clusters, that expand from subclonal in primary samples to clonal in at least one metastatic sample from all 20 patients with primary and metastatic samples calculated using Ingenuity Pathway Analysis. **f** As for **e** but showing enrichment for gene networks associated with metastatic-clonal genes. **g** As for **e** but with metastatic-clonal genes in dural and parenchymal metastases analyzed separately. **h** As for **g** but examining gene sets defined by their upstream regulator. Enrichment was calculated using two-sided Fisher's exact test. Plotted *P*-values are unadjusted. Source data are provided as a Source Data file.

into the brain parenchyma. Given that we observed both dural and parenchymal metastases in our cohort, we asked if there were any genetic differences between these tumors, positing that parenchymal metastases may have gained additional driver mutations as compared to the dural metastases. We investigated the genes with mutations in clusters which became clonal in the metastases from the 20 patients from whom we had primary and metastatic samples. These genes were highly enriched for factors involved in calcium signaling including mutations affecting the EP300 paralog *CREBP* (Fig. 3e, Supplementary Fig. 10a). We also found enrichment for genes involved in the FAT10 cancer signaling pathway (Fig. 3e, Supplementary Fig. 10b).

We further explored these metastatic-clonal genes and found that they were significantly associated with networks related to processes including DNA replication, recombination and repair, and neurological disease (Fig. 3f). We then performed the same comparison, after dividing the metastatic-clonal genes based on the dural or parenchymal metastatic location as an exploratory analysis to dissect the mechanisms which are most frequently affected by genetic alterations in the different metastatic locations. This comparison was carried out on a limited number of samples (nine dural, ten parenchymal) in order to explore the possibility of any differences between the two sites.

Following correction for multiple testing, metastatic-clonal genes in the parenchymal metastases showed enrichment for genes involved in calcium signaling and 'regulation of EMT in development' ($q = 0.013$ and $q = 0.02$, respectively). Metastatic-clonal genes in the dural metastases, however, were enriched for factors associated with aldosterone signaling in epithelial cells ($q = 0.047$) (Fig. 3g). Still considering the specific metastatic site (dural or parenchymal), we asked whether these metastatic-clonal genes fell under the influence of a particular upstream regulator. Following correction for multiple testing we found no enrichment for genes under a particular regulator being altered in the parenchymal metastases, but observed a trend towards enrichment for genes regulated by the long noncoding RNA mir17HG ($q = 0.07$). Similarly, in the dural metastases we observed a trend towards enrichment for genes regulated by the polycomb repressive complex subunit EZH2 ($q = 0.07$), a well described epigenetic driver of prostate cancer[21,22] (Fig. 3h). While the nature of these interactions remains to be defined, these data suggest particular mechanisms which may drive the metastasis of prostate cancer to the dura or parenchyma, which may be regulated by epigenetic or tumor microenvironmental factors that were not assessed.

These data show the heterogeneity present in PCBM and support a model of clonal selection in the metastatic setting, along with continued clonal evolution occurring in the primary tumor after metastasis has occurred. While further data are needed better to understand the different mechanisms driving dural or parenchymal metastasis, the enrichment for calcium and FAT10 signaling in these metastases in general suggest possible mechanisms driving metastasis to the brain. Furthermore, the affected networks reiterate the potential importance of altered DNA damage repair pathways in PCBM.

## Discussion

As improvements in prostate cancer survival have allowed an increase in metastases to less common metastatic sites, we investigated the genetic landscape of PCBM in comparison to non-brain metastases, to better understand the processes associated with this particular disease progression.

We observed an increased level of somatic alterations in the PCBM compared to the CRPC500 cohort of non-brain prostate cancer metastases, along with an increased frequency of PROfound-gene alteration, with biallelic inactivation of at least one of these genes detected in 8/51 (15.7%) patients. Given this, we compared the presence of the HRD mutational signature in the PCBM and CRPC500 cohorts. Importantly, and in line with a recent study on the mutational signatures present in brain metastases from colorectal cancer[14], we observed a significant enrichment for the mutational signature of HRD (as detected by SigMA) in PCBM compared to non-brain metastases. Moreover, a significant enrichment of the HRD signature was also present in the primaries of PCBM compared to a high-grade subset of PCA in the TCGA-cohort. Given its enrichment in the primaries of PCBM compared to TCGA, the presence of HRD signature may

serve as a useful risk stratification biomarker for metastatic PCa progression to the brain. It is important to note that, in 24/51 patients, no matched normal tissue was available, so mutations were called against a pooled normal. However, steps were taken to prevent inflating the number of mutations, and results retained significance when considering only cases for which matched normal tissue was available.

The PROfound phase 3 trial demonstrated prolonged overall survival in the cohort of patients with mCRPC and pathogenic alterations affecting *BRCA1*, *BRCA2*, or *ATM* as well as benefits in the overall trial population with alterations in any of the PROfound genes when treated with the PARP inhibitor olaparib after progression on enzalutamide or abiraterone. However, while PARPi, such as Olaparib[23,24] or niraparib[25], have been shown to be active on brain metastases in patients with HRD breast cancer and in murine models of brain metastases[26,27], patients with known brain metastases were excluded from the PROfound trial[18]. We detected pathogenic alterations meeting the criteria for the PROfound trial in 10/51 (19.6%) patients in the PCBM cohort, with five (9.8%) patients harboring qualifying alterations in *BRCA1* or *BRCA2*.

This study, therefore, suggests that a relevant proportion of men with PCBM may benefit from PARPi given the high frequency of HRR alterations, the prevalence of the HRD signature and the presence of pathogenic molecular events in the PROfound genes.

The 20 cases for which primary and metastatic samples were available allowed us to investigate the clonal evolution occurring in PCBM, while the inclusion of patients with both dural and parenchymal metastases allowed comparison of the processes which may drive metastases to these specific sites. In keeping with a model of selection of particular subclones in the metastatic niche[28] we observed both heterogeneity in the different regions and clonal expansion in PCBM, particularly involving genes related to calcium signaling and FAT10 signaling. The genes associated with calcium signaling included EP300 paralog CREBP which regulates genes involved in cell growth in prostate cancer[29], while those associated with the FAT10 pathway included genes such as the high mobility group gene *TCF7L2* and *CTNNB1*, known to regulate processes such as cell migration and metastasis[30]. While the genes with mutations which became clonal in the dural metastases were enriched for molecules involved in aldosterone signaling, and possibly regulation by EZH2, we observed enrichment in parenchymal brain metastases for genes involved in the same calcium signaling pathway as was enriched in the set of all metastatic-clonal genes, along with genes involved in developmental EMT. While these data are not conclusive, they suggest the importance of alterations to these pathways in parenchymal brain metastases. The enrichment for alterations affecting mir17HG is also a potential link between parenchymal metastases and HRD as, in concert with SIRT1, this long noncoding RNA has been suggested to promote double-strand DNA break repair[31]. This, along with the putative enrichment for genes regulated by EZH2 may, alternatively, suggest that epigenetic or microenvironmental alterations could also explain this propensity.

These observations may serve as a starting point for more in-depth investigation into the processes driving metastasis to the brain in particular. Furthermore, the enrichment for the clonal expansion of mutations affecting genes involved in DNA repair, highlights the association between defective DNA repair and PCBM, identified by our analysis of the mutational signatures prevalent in PCBM cohort, and the frequency of alterations affecting HRR genes.

Taken together, the enrichment for HRD mutational signature and for HRR alteration (HRD mutational signature and

PROfound-gene alterations) in PCBM compared to non-brain PCa metastases suggests that HRD is a salient characteristic of PCBM. In addition, given its enrichment in the primaries of PCBM compared to TCGA, the HRD signature may be a useful risk stratification biomarker for metastatic PCa progression to the brain. Finally, the detection of well documented pathogenic events in HRR in a significant subset of PCBM cohort and together with the benefits patients with such alterations shown to Olaparib therapy in PROfound, would support a potential role for PARP-inhibitor therapy in a population of men who were not included in the PROfound clinical trial.

## Methods

**Patient selection and tumor procurement.** Tumor samples were collected from Pathology Departments in university and Cantonal Hospitals across Switzerland (Institute of Pathology, Bern/ Institute of Neuropathology, Zurich/ Institute of Medical Genetics and Pathology, Basel/ Institute of Pathology, Aarau/ Institute of pathology, Münsterlingen/ Institute of Pathology, Liestal/ Institute of Pathology, St. Gallen) and from the Department of Pathology and Laboratory Medicine, and Urology, Emory University School of Medicine, Atlanta, USA. Inclusion criteria were defined as patients having available formalin-fixed paraffin-embedded (FFPE) blocks from confirmed CNS or meningeal metastases of prostate carcinoma and, if available, from the matched primary tumor and normal tissue (Supplementary Table 1). All analyses were carried out in accordance with protocols approved by the Ethical Committee Bern (Project ID: 2019–00328). No participant compensation was applied for the current study.

**Study population.** Our cohort includes samples from 51 patients. Patients qualified for inclusion in this study if a written consent or no documented refusal was available (Human Research Act, HRA, Swiss Confederation; Art. 34). We collected archived FFPE tissue from CNS (brain/spinal cord) and meningeal metastases with matched primary tumors in 20 cases. Most tumor samples corresponded to diagnostic biopsies (from prostate or CNS/dura), transurethral resections (TURP) or prostatectomy specimens. Primary tumors and metastases from patients P1, P32, P43-46, P48 and P49 were taken from autopsy tissue, with patients P1 and P43 harboring additional diagnostic biopsies. At least one metastatic sample was included from 51/51 (100%), with patient 43 harboring metastatic samples at different time-points. Additionally, from 20/51 (39.2%) patients primary tumor tissue was available, with 6/10 patients (P1, P6, P9, P29, P42, and P44) including primary tumor samples at different time-points. In total we selected 168 tumor areas, 63 from primary tumors and 105 from metastases. IHC was conducted on 149/168 of the total areas (88.7%), including 61/67 (97%) primary tumor and 88/105 (84%) metastasis areas. All 168 selected areas underwent both molecular analyses (i.e., WES and targeted RNA) and data were obtained successfully in 168/168 (100%) of the cases (Supplementary Fig. 1).

**Pathology review.** All tissue slides (HE and IHC) were scanned and uploaded in CaseCenter (http://ngtma.path.unibe.ch/casecenter/). Through the digital microscope application CaseViewer, the slides were reviewed and annotated for further core biopsy punching. Morphological and immunohistochemical assessment was done by ARC and supervised by a board-certified pathologist (MAR). Based on the morphology, we assessed the presence of different Gleason patterns, intraductal carcinoma, ductal histology, and neuroendocrine differentiation. For each specimen, we selected representative blocks to best recapitulate the heterogeneity of the above features. IHC stains were performed on all selected blocks after first review. Cases with limited amount of tissue were assessed only morphologically. For each case, p53 (clone DO-7; Dako-Agilent; 1:800) PTEN (clone 6H2.1; Cascade Bioscience; 1:400) and ERG (clone EP111; Dako-Agilent; 1:50) were stained. Additionally, if neuroendocrine features were present, Chromogranin-A (clone DAK-A3; Dako-Agilent; 1:1600), Synaptophysin (clone 27G12; BioSystems; 1:100) and PSA (polyclonal; Dako-Agilent; 1:4000) were added, as were CK5/6 (clone D5/16B4; Merck; 1:4000) and p63 (clone 7JUL; BioSystems; 1:40) for suspected intraductal carcinoma. Finally, by combining morphological and immunohistochemical features, we identified and selected up to three heterogeneous Regions of interest (ROIs) within primary tumors and metastases. In cases showing homogeneous morphological and immunohistochemical features throughout all examined slides, up to three tumor areas were randomly selected.

**Core biopsies for genomics and transcriptomics analyses.** Within the selected ROIs, core biopsies (each 1 mm diameter) were annotated, punched, and separately used in order of priority for targeted RNA and WES.

**DNA extraction and whole-exome sequencing.** After deparaffinization, DNA was extracted from selected FFPE core biopsies (1 mm diameter) of matched tumor and normal tissue using the QIAamp DNA micro kit (Qiagen). Quality and quantity were determined by real-time PCR (Agilent NGS FFPE QC Kit).

10–200 ng of DNA underwent library preparation and exome capture using the SureSelect$^{XT}$ low input protocol with Human All Exon V7 (Agilent) as per manufacturer's guidelines. Multiplexed libraries were sequenced on an Illumina NovaSeq 6000 (2 × 100 bp) at the Clinical Genomics Lab Inselspital Bern University Hospital (Supplementary Table 4 and Supplementary Data 4).

**Targeted RNA extraction and sequencing.** Selected FFPE core biopsies (1 mm diameter) of tumor tissue were subjected to DNA and RNA extraction using the AllPrep DNA/RNA FFPE kit (Qiagen). Concentrations were determined with a Qubit 2.0 fluorometer (Life Technologies). 15–20 ng of RNA were reverse transcribed to cDNA (Superscript VILO, Invitrogen). cDNA and 10 ng of DNA were used for library preparation with the Ion AmpliSeq Library Kit Plus with a prostate specific custom multiplex RNA[32] panel and barcode incorporation (Ion Torrent, Thermo Fisher). Template preparation of the multiplexed libraries was performed on the Ion Chef system with subsequent sequencing on the Ion S5 XL sequencer (Ion Torrent, Thermo Fisher).

For gene-fusion analysis for each sample all the reads that completely cover the fusion genes specific amplicons from one end to the other end (end-to-end reads) were collected and processed in order to detect the presence of fusions. Fusion genes were called when filtering criteria were met, based on the percentage of the specific end-to-end reads, the number of breakpoint reads and the presence of possible bias toward the forward of the reverse end-to-end reads (Supplementary Table 4; Supplementary Data 5).

**Microsatellites analysis.** By using extracted DNA as mentioned above, we analyzed six microsatellite loci (BAT25, D17S250, BAT26, BAT40, D5S346, D2S123). Fluorescent primers were used[11] and the analysis was performed by capillary electrophoresis using the ABI 3500 genetic analyzer (Applied Biosystems).

**Sequence data processing pipeline and single nucleotide variant identification.** Reads obtained were aligned to the reference human genome GRCh38 using Burrows-Wheeler Aligner (BWA, v0.7.12)[33]. Local realignment, duplicate removal, and base quality adjustment were performed using the Genome Analysis Toolkit (GATK, v4.1) and Picard (http://broadinstitute.github.io/picard/). Somatic single nucleotide variants (SNVs) and small insertions and deletions (indels) were detected using Mutect2 (GATK 4.1.4.1)[34] and Strelka2 v2.9.10[35]. Only variants detected by both methods were reported. We filtered out SNVs and indels outside the target regions (i.e. exons), those with a variant allelic fraction (VAF) of <5% and/or supported by <3 reads. We excluded variants for which the tumor VAF was <5 times that of the paired non-tumor VAF, as well as those found at >5% global minor allele frequency of dbSNP (build 137). We further excluded variants identified in at least two of a panel of 210 non-tumor samples, including the non-tumor samples included in the current study, captured and sequenced using the same protocols using the artifact detection mode of Mutect2 implemented in GATK. For samples for which we had no matched normal tissue, we also removed variants present at VAF of > 0.1% in the ExAC non-TCGA database of normal germline samples. All indels were manually inspected using the Integrative Genomics Viewer[36]. Hotspot missense mutations were annotated using the published resources[37,38].

We carried out two analyses of the effect of differences in sequencing depth between cohorts used in this study, by adjusting the thresholds for alternate allele reads, and by simulating downsampling of our sequencing data. The minimum of one supporting read in TCGA (sequenced to ~100X) is equivalent to ~2.6 reads at our sequencing depth. Similarly, the minimum of ten supporting reads in CRPC500 (sequenced to ~160X) is equivalent to ~16 reads at our sequencing depth. We, therefore, carried out an analysis of our variants using the most stringent combination of thresholds used across these cohorts (i.e., VAF 5% and supported by at least 16 reads) and compared the observed TMB between studies.

To examine the effect of overall sequencing depth on variant calling, we performed a second analysis to simulate the effect of downsampling of our sequencing data to depths (~258X) comparable to TCGA and CRPC500 (the lower being ~100X). Given our current VAF (5%) and supporting read (>3) thresholds, these thresholds would have to be multiplied by 2.6 (~258X/~100X) to simulate the effect of downsampling our ~258X PCBM data to ~100X. This resulted in new thresholds of 12.9% VAF, and a depth threshold of eight alternate allele reads which were used to filter called variants before repeating the comparison of TMB between studies. Data were converted to maf format using maftools and visualized using ggplot2 and ComplexHeatmap in R version 3.6.1. TCGA data were downloaded using TCGAbiolinks.

**Allele-specific copy number analysis.** Allele-specific copy number alterations were identified using FACETS (v0.5.6)[39], which performs joint segmentation of the total and allelic copy ratios and infers purity, ploidy, and allele-specific copy number states. Copy number states were collapsed to the gene level using the median values to coding gene resolution based on all coding genes retrieved from the Ensembl (release GRCh38).

Genes with total copy number greater than gene-level median ploidy were considered gains; greater than ploidy + 4, amplifications; less than ploidy, losses; and total copy number of 0, homozygous deletions. Somatic mutations associated

with the loss of the wild-type allele (i.e., loss of heterozygosity [LOH]) were identified as those where the lesser (minor) copy number state at the locus was 0. For chromosome X, the log ratio relative to ploidy was used to call deletions, loss, gains, and amplifications. All mutations on chromosome X in male patients were considered to be associated with LOH[40]. Data were visualized using ComplexHeatmap in R version 3.6.1.

**Mutational signature and HRD**. Decomposition of mutational signatures was performed using deconstructSigs[41] and MutationalPatterns[42] based on the set of 60 mutational signatures ("signatures.exome.cosmic.v3.may2019")[43,44], for samples with at least 20 somatic mutations. Results from both methods were averaged. To increase robustness when running deconstructSigs, the mutations for each sample were bootstrapped 100 times and the mean weights across these 100 iterations were used. Sensitive calling of SBS3 (HRD) alone was subsequently performed using SigMA[17], which uses a multivariate analysis to detect the presence of SBS3. The score from this was used in comparisons between datasets, along with a thresholded call of signature presence based on the 'pass_mva_strict' annotation from this analysis. Data were visualized using ggplot2 and ComplexHeatmap in R version 3.6.1.

**Clonality of single nucleotide variants**. Clonal prevalence analysis was conducted using the hierarchical Bayesian model PyClone, and the ABSOLUTE V2.0 algorithm in the case of samples used in analysis of clonal evolution. PyClone estimates the cellular prevalence of mutations in deeply sequenced samples, using allelic counts, and infers clonal structure by clustering these mutations into groups with co-varying cellular frequency. PyClone was run using a two-pass approach, whereby mutations whose cellular prevalence estimate had standard deviation >0.3 were removed before a second pass analysis was run. A cellular prevalence of >80% was used as a threshold for clonality. ABSOLUTE infers CCF from the reads supporting the reference/alternative allele, in conjunction with segmented copy-number data from WES, and was run after patching as described here: https://github.com/broadinstitute/PhylogicNDT/issues/4#issuecomment-555588341. Solutions from ABSOLUTE were manually curated to assure the solution matched the ploidy estimate generated by FACETS.

**Phylogenetic analysis**. CCF histograms generated by ABSOLUTE were used as the input to PhylogicNDT[45] to find clusters of mutations, infer subclonal populations of cells and their phylogenetic relationships, and determine the order of occurrence of clonal driver events. PhylogicNDT was run using the parameters "Cluster -rb -ni 1000" to cluster and build the phylogenetic tree with 1000 iterations. Data were visualized using ggplot2 in R version 3.6.1.

**Ingenuity pathway analysis**. Metastatic-clonal genes were determined for each patient as those in clusters which were subclonal in the primary, but clonal in the metastatic samples. The lists of all such genes from each patient were combined and significantly altered pathways, networks, or genes under the same upstream regulator, were determined from this list using the Ingenuity Pathway Analysis (IPA) program as previously described[46]. For the analysis, metastatic-clonal genes, were mapped to networks available in the Ingenuity database and ranked by a score indicating the likelihood of finding those genes together by chance. The two most enriched pathways and four most enriched networks are shown. The same analysis was performed separately on metastatic-clonal genes in dural and parenchymal metastases. Data were visualized using ggplot2 in R version 3.6.1.

**Reporting summary**. Further information on research design is available in the Nature Research Reporting Summary linked to this article.

## Data availability

Whole-exome sequencing data generated in this study are available on EGA [study ID: EGAS00001005091], access can be obtained by contacting corresponding authors Mark A. Rubin. The TCGA and CRPC500 publicly available data used in this study are available on cBioPortal [TGCA: https://www.cbioportal.org/study/summary?id=prad_tcga; CRPC500: https://www.cbioportal.org/study/summary?id=prad_su2c_2019]. The remaining data are available within the Article, Supplementary Information or Source Data file.

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

## Acknowledgements

The authors would like to thank Mariana Ricca at the University of Bern for her expert assistance in editing and preparing the manuscript for submission and to the Translational Research Unit (TRU) of the Institute of Pathology (University of Bern) for the collaboration and support throughout the pathology review. This study was supported by an NCI (NIH) grant P50 CA211024 (S.A.T. and M.A.R.); Swiss Personalized Health Network grant SOCIBP (H.M., M.A.R.), and the Swiss Cancer League (C.K.Y.N., S.P., and M.A.R.). S.P. is supported by The Prof. Dr. Max Cloëtta foundation.

## Author contributions

A.R.-C., J.G., S.G., S.P., and M.A.R. designed the study and the experiments. A.R.-C. and J.G. performed experiments and analysis of the results. J.G. performed the bioinformatic analysis. A.R.-C., J.G., D.A., A.F., S.G., S.P., and M.A.R. developed the concept. A.R.-C. and M.A.R performed the pathology review and immunohistochemical evaluation. D.A. and S.G. provided a clinical perspective to the results. S.M., U.A. and V.P performed and coordinated the DNA and RNA-sequencing. E.V. conducted the MSI-status analyses. J.C., A.R.-C., and M.A.R. developed the pathology review approach. A.G. and C.K.Y.N. developed modules for the bioinformatic pipeline and provided bioinformatic support. S.A.T. developed the DNA and RNA probe set design for prostate cancer. E.H., V.G., Ac.F, E.J.R., R.G., I.F., W.J, G.C., A.O.O., L.B., H.M., and G.T. provided material and/or clinical data. M.A.R. provided administrative, technical and material support. A.R.-C., J.G., D.A., A.F., S.G., S.P., and M.A.R wrote the initial draft of the manuscript and all authors contributed to the final version.

## Competing interests

S.A.T. and M.A.R. are co-authors (and included in the royalty streams) on the patent US7718369B2 issued to the University of Michigan and the Brigham and Women's Hospital, on ETS gene fusions that have been licensed to Hologic/Gen-Probe Inc., who sublicensed rights to Roche/Ventana Medical Systems, and LynxDX. S.A.T. has served as a consultant for and received honoraria from Janssen, and Astellas/Medivation. S.A.T. has sponsored research agreements with Astellas/Medivation. S.A.T. is a cofounder of, prior consultant for, equity holder in, and employee of Strata Oncology. S.G. plays a consulting or advisory role to Astellas Pharma (Inst), Curevac (Inst), Novartis (Inst), Active Biotech (Inst), Bristol-Myers Squibb (Inst), Ferring (Inst), MaxiVax, Advanced Accelerator Applications, Roche, Janssen (Inst), Innocrin Pharma (Inst), Sanofi, Bayer (Inst), Orion Pharma GmbH, Clovis Oncology (Inst), Menarini Silicon Biosystems (Inst), MSD (Inst). S.G. is a co-author of the patent Method for biomarker (WO 3752009138392 A1). S.G. is an honorary member of Janssen and has also ties to Nektar, ProteoMediX. M.A.R. is on the Scientific Advisory Board of NeoGenomics, inc. C.K.Y.N. serves as a consultant for Repare Therapeutics USA. L.B. has served as an advisor for Janssen, Bayer, and Roche, has received honoraria from Janssen, and has sponsored research agreement with Novartis. The remaining authors declare no competing interests.
