## [Peer Review File · Nature Communications]

Alterations in Homologous Recombination Repair Genes in Prostate Cancer Brain MetastasesReviewers' Comments:

Reviewer #1:

Remarks to the Author:

The authors describe the genomics of a rare complication of metastatic prostate cancer: brain metastases based on the analysis of multiple regions of both the metastasis and the corresponding primary tumors. The analyses were well described and represent a novel finding albeit on a very rare complication of metastatic prostate cancer.

I have some problems with the conclusion that NF1 and RICTOR mutations are enriched in brain metastasis and mainly the possible causative relation as described in the abstract. In many tumor types the most aggressive cancers cause brain metastases and there is no control group in the analysis correcting for just "poor prognostic" metastatic tumors and no cases of brain specific NF1/RICTOR mutations have been described. Thus these may just be markers of poor prognosis in prostate cancer.

specific comments:

1) The text of the results section is very elaborate and often meanders between lengthy descriptions of the data and description of individual cases. I would suggest the text could be shortened. eg p5 the data on IHC analysis is not really informative so could be placed in the supplemental methods. P6 where the differences are denoted in general features could easily be a table. p7 the description of the signatures is lengthy and not really clear.

2) p7-8 authors suggest that PCBM can be "distinguished" by signatures. Based on the presented data this is not true, indeed there is enrichment and loss of certain signatures, however this is insufficient to distinguish the PCBM based on this.

3) p8-9 the analysis on PCBM specific mutations should be complemented with some more clinical characteristics such as pre-treatment history. It may well be that NF1 and RICTOR mutations arise as part of therapy resistance mechanisms not specific for PCBM (or as stated above as poor prognostic markers).

4) p11 the text on the analysis of NF1 is difficult to follow. Please clarify as I cannot reconstruct the numbers...

5) the clonal selection analysis seems to be somewhat disconnected from the rest of the paper. Although in itself the analysis seems to have been performed in a proper manner the conclusion that the ability to metastasize to the brain is an early event is odd. With only 2 patients analyzed in this manner and no conclusive brain metastasis driver genes, this conclusion seems overstated. The analysis, in my opinion, does not seem to contribute to the overall message of the paper.

Overall the conclusion that there seems to be an enrichment in the HRD deficient tumors of PCBM can be viewed as the most important conclusion. An analysis aggregating the findings: NF1/RICTOR and HRD would complement the manuscript, are they mutually exclusive or overlapping?

Reviewer #2:

Remarks to the Author:

The manuscript by Rodriguez et al., "The Genomic Landscape of Prostate Cancer Brain Metastases" describes the genetic analysis of 28 prostate cancer cases with brain metastases (PCBM) using WES and targeted RNA sequencing, and focuses on the identification of distinct alterations enriched in the PCBM cohort by comparing to the Stand Up to Cancer/PCF CRPC500 cohort (416 prostate cancer patients with non-brain metastases). Matched primary PCa from 10 patients were also analyzed. Since

brain metastasis rarely occurs in prostate cancer, it's appreciated that this study assembled a cohort of these uncommon samples.

Here, the authors highlight frequent aberrations in NF1 (and RICTOR) and signature of homologous repair defects in brain mets that they claim represent potential precision therapeutic avenues. Below are some aspects that should be addressed.

Major concerns:

1. FFPE samples are often sub-optimal and resulting NGS-based variant detections are prone to false positives. Therefore, it is curious to find that both PCBM primary tumors and brain metastases had significantly higher levels of mutations than the TCGA PRAD and the CRPC500 cohorts, when these two publically available cohorts are composed of a high ratio (if not all) of frozen tissues. It's not obvious from the manuscript that if the following factors have been considered. (1) Enzymatic repair of FFPE DNA can help to reduce false mutation frequencies by removing artefactual C-T conversions that result from cytosine deamination during formalin fixation. Was the step included in the sample preparation process? (2) The authors set a threshold to filter out those SNVs and indels with a variant allelic fraction (VAF) of <1% and/or those supported by <3 reads. Can the settings efficiently eliminate potential FFPE artifacts that are prone to low allele frequency? False discovery assessments according to variant allele frequency should be implemented. (3) A complete list of mutation calls with detailed information and VAF should be provided. (4) Were the TCGA and CRPC500 cohort reanalyzed by the same pipeline? Data should be treated the same way for direct comparison. (5) Are matched normal samples sequenced for each patient? Tumor-alone sequencing could add another level of noise in mutational analysis.
2. Clinical significance of brain-mets as a unit of analysis is not clear. Do prostate cancers that are positive for brain-mets show a different metastatic profile for other sites? Do these show more aggressive clinical course than cases with non-brain-mets, different survival, different histomorphology, or different immune infiltration profile?
3. The cohort size of brain-mets analyzed here, 28 metastases/ 10 matched PCa- is higher than any previous studies- but then no previous study purported to describe brain-met specific genomic aberrations. Here the cohort size prompts concern for incidental over-representation of some aberrations as brain-met specific association. The authors need to support their NF1/ RAS pathway association with functional analysis, since data from prostate brain-mets is fairly scant in literature.
4. A comparator missing from the study is patient matched non-brain mets. This would have helped compare/contrast if tumor cell clones establishing as brain-mets are qualitatively distinct from non-brain-mets. Considering that the authors have access to brain-mets from rapid autopsy program- wherever brain-mets are obtained from, those cases do certainly have metastatic tissues in other sites as well, in much higher abundance if anything. A direct comparison of brain- and non-brain- mets from the same cases could help directly identify any brain-met specific aberrations.
5. The authors need to distinguish the possibility that in some cases, prostate-mets tend to preferentially metastasize to brain, rather than other organ sites; from the other, simpler possibility that presence of brain-mets are merely indicative of a higher metastasis burden: more mets, go to more places. If the former possibility be supported by data, analysis of brain-met specific aberrations would be moot. However, as seen in Figure 1 and elsewhere, arguably, mCRPCs that are seen to display a higher mutation burden, are presumably associated with a higher burden of metastasis sites- including as far as brain. The authors need to rule out this possibility to rationalize a brain-met specific hypothesis.
6. The possibility that PCBMs are simply associated with higher "mets-burden", makes comparison of PCBMs with non-brain mets across public data less attractive- as many of those cases may simply represent less-disseminated mets cohorts.
7. The 28 PCBM cases included here- what is the burden of non-brain mets in this case. Do these cases show any distinct pattern of metastatic sites compared to cases without PCBMs?
8. Please show mutation burden (Figure 1, S1.2, etc.) in the more standard format of "number of mutations/ Mb sequencing data". This is important to normalize the data across the dataset, subject to variations in coverage, as well as outside data, in public domain/ publications. Figure 1b, the y-axis scale of the one extreme outlier PCA sample can be broken, to help scale the rest of the graph more appropriately.

9. "The mismatch repair-defective PCA (P1), harboring an MLH1 somatic missense mutation (p.Arg522Trp) coupled with the copy number loss of the other allele, with mutation rates ranging from 964 to 1820 mutations/sample (Figure 1a)". Can the authors confirm if this sample shows microsatellite instability (MSI) signature?? Incidentally, p.Arg522Trp is annotated in ClinVar as a germline variant of unknown significance; in COSMIC this mutation is noted only in one case of CRC with unknown status of MSI. The mutational signatures shown in Figure 1a doesn't seem to be consistent with MMRD for this patient, either.

10. The mutations included in Figure 2 and 3: please provide a detailed list of specific mutations with nucleotide/ amino-acid changes showing specific aberrations, along with concomitant status of copy number or copy neutral LOH.

11. SCNA data- Figure S2.3- Can this be parsed to highlight SCNAs enriched in the PCBM? Figure S2.2 a and b figure labels are identical- need to fix.

12. In this study, it seems that all mutations are viewed without the assessment of "driver vs. passenger" or "biological significant vs. variant of unknown significance". A good example is RICTOR. Mutations of this gene were reported to be significantly enriched in the PCBM cohort. Hence, possible activation of the druggable pathway PI3K/AKT/mTOR was indicated. However, RICTOR is not an oncogene with well-known hotspot activating mutations. In addition, positions of RICTOR mutations were not actually provided in the manuscript for evaluation. More evidence needs to be provided to call clinical relevance of RICTOR mutations; ie: Page 11: "observed mutations affecting RICTOR (100% missense)"- please provide the details of the mutations to allow an evaluation if these are likely functionally relevant or not.

13. In Materials and Methods and elsewhere- no indication of assessment of LOH by copy neutral allelic imbalance was noted. Related to that, this study does not distinguish monoallelic- and biallelic- inactivation of tumor suppressor genes. Assessments of tumor suppressors (such as NF1, PTEN, TP53, RB1, BRCA) based on "one-hit" events (heterozygous loss or mutation without second hit) are not meaningful for this class of genes.

14. In the highlighted PROFOUND clinical trial (de Bono, J. et al. 2020, Olaparib for Metastatic Castration-Resistant Prostate Cancer. NEJM 382, 2091-2102), a key observation seems to be the rather insignificant responsiveness of non-BRCA HR mutants to PARP inhibitor therapy. It doesn't really support the inclusion of the wider net of HR gene aberrations as proposed here- please comment/ revise the text.

15. The proposed equivalence of NF1 in PCBM with YAP1 mutations in lung cancer is highly speculative- YAP1 is well characterized as a regulator of Hippo pathway. The one cited cross-talk with RAS pathway is likely context specific, not its central signaling axis.

Minor Points:

1. In Supplementary Figure 1, 4/24 should be 4/28.

2. "PIK3CD" is misspelt in a few places as "PI3KCD".

NCNCOMMS-20-18280-T: Point by Point Response to Reviewers' Comments

Reviewer #1

The authors describe the genomics of a rare complication of metastatic prostate cancer: brain metastases based on the analysis of multiple regions of both the metastasis and the corresponding primary tumors. The analyses were well described and represent a novel finding albeit on a very rare complication of metastatic prostate cancer.

I have some problems with the conclusion that NF1 and RICTOR mutations are enriched in brain metastasis and mainly the possible causative relation as described in the abstract. In many tumor types the most aggressive cancers cause brain metastases and there is no control group in the analysis correcting for just "poor prognostic" metastatic tumors and no cases of brain specific NF1/RICTOR mutations have been described. Thus these may just be markers of poor prognosis in prostate cancer.

Specific comments:

1) The text of the results section is very elaborate and often meanders between lengthy descriptions of the data and description of individual cases. I would suggest the text could be shortened. eg
 P5 the data on IHC analysis is not really informative so could be placed in the supplemental methods.
 P6 where the differences are denoted in general features could easily be a table.
 P7 the description of the signatures is lengthy and not really clear.

Response #1

We thank the Reviewer for these editorial comments. We have adjusted the manuscript by now including the IHC and SNV/InDels information in tables Supplementary data SD2 and SD4, respectively.

New SD2. Number of patients with pathological protein expression by IHC

	ERG expression	p53 aberrant expression (loss or >50% positive cells)	PTEN cytoplasmic loss	PTEN nuclear loss
Primary tumors (N=10)*	4 (40%)	8 (80%)	7 (77.9%)**	9 (100%)**
Metastases (N=24)*	13 (54.2%)	20 (83.3%)	18 (78.3%)**	23 (100%)**

*IHC was conducted in 10/10 primary tumors and in 24/28 metastases; (s. Methods).

New SD4. Means and differences of non-synonymous mutations (SNVs and InDels) within PCBM, CRPC500 and TCGA-cohort (mutations/Mb)

PCBM metastases vs CRPC500				PCBM primaries vs TCGA			
	PCBM metastases mean*	CRPC500 mean*	Difference in means*		PCBM primaries mean*	TCGA mean*	Difference in means*
SNV	6.1	1.02	5.08	SNV	4.79	0.55	4.24
Insertions	0.2	0.02	0.18	Insertions	0.17	0.026	0.14
Deletions	0.34	0.06	0.28	Deletions	0.34	0.03	0.31
*mutations/Mb				*mutations/Mb			

Changes to manuscript #1

We have added **Supplementary data SD2** and **SD4** and shortened the text on page 5 and page 6. We rewrote the signatures section on page 6-7.

The text now reads:

Page 5

“We further performed IHC analysis of proteins with frequently altered expression in PCa (e.g. ERG, p53 and PTEN); (s. **Supplementary Data SD2 and SD3**).”

Page 6

“Metastatic samples from the PCBM cohort had significantly higher levels of non-synonymous single nucleotide variants (SNV), insertions and deletions than non-PCBM metastatic PCa samples ($q < 0.01$, Wilcoxon test). Primary samples from the PCBM cohort also had significantly higher levels of SNVs, insertions and deletions than the TCGA primary cohort. ($q < 0.01$, Wilcoxon test); (**Supplementary Data SD4**).”

Page 6-7

“Mutational signatures were determined using a previously described approach¹⁵ (see methods for details). We report signatures with $>15\%$ contribution in at least one tumor sample. As expected in a population of older men (median age 71 years at time of PCBM diagnosis), the most common mutational signature was SBS1 – deamination of 5-methylcytosine, a signature associated with ageing¹⁶. The signature for defective homologous repair (SBS3) was observed in the metastases of two patients (P9 and P30), while it was absent in the matched primary samples from these patients¹⁶ (Figure 1a).

Substantial differences in mutational signatures were detected between primary and metastatic samples. For example, the matched primary PCa in patient P29 predominantly presented signatures for defective nucleotide excision repair (NER; SBS30), polymerase epsilon exonuclease domain mutations (SBS10b) and NER activity (SBS32) (Figure 1c). In the PCBM sample from this patient, the representation of these signatures decreased as 16 different signatures were detected, compared to four and nine in the primary tumor samples. In patient P6, from whom samples were collected over seven years the prevalence of SBS1 (ageing) reduced while SBS16 (transcriptional strand bias of NER activity) increased progressively throughout the course of disease progression. Similarly, SBS9 was detectable only in the latter five samples from this patient (Figure 1a).

Six mutational signatures were found to have significantly greater representation in the PCBM compared to the CRPC500 cohort (e.g. SBS5, SBS12, SBS16, SBS32, SBS37, SBS39) ($P < 0.0001$, T-test) (Figure 1d). Two of these signatures, SBS37 and SBS39, are of unknown etiology but have been detected in both prostate cancer and cancers of the CNS¹⁶. While their specific cause is unknown, SBS12, SBS16 and SBS32 have been linked to transcriptional strand bias of transcription-coupled nucleotide excision repair (NER), while SBS5 is associated with ageing, as well as mutations in the transcription-coupled NER gene ERCC2”

2) p7-8 authors suggest that PCBM can be "distinguished" by signatures. Based on the presented data this is not true, indeed there is enrichment and loss of certain signatures, however this is insufficient to distinguish the PCBMs based on this.

Response #2

We agree that our data are not sufficient to conclude that PCBM were distinguished by their mutational signatures. We have extended this analysis by clustering the samples from the PCBM and CRPC500 cohorts by their mutational signatures. This revealed two main clusters, one of which was highly enriched for the PCBM samples.

New Supplementary Figure S1.3 (related to Fig. 1). Unsupervised hierarchical clustering of PCBM and CRPC500 metastatic samples. Signatures showing greater than 15% representation in at least 1 sample in the PCBM and CRPC500 cohorts were used for unsupervised hierarchical clustering using the Ward D2 algorithm. Dataset and site for each sample are shown in the top annotations. Cluster assignment (1 and 2) is shown in the bottom annotation.

Changes to manuscript #

We have modified the text on page 7 regarding our conclusions about the mutational signatures and have included the figure above as **Supplementary Figure S1.3** showing the clustering of samples by the prevalence of signatures. The text on page 7 now reads:

“We performed unsupervised hierarchical clustering on the PCBM and CRPC500 metastatic samples, using the fraction of mutations assigned to each signature. This resulted in two main clusters of samples, one of which was highly enriched for the PCBM samples ($q = 1.37e-14$, Fisher’s exact test), while no enrichment was detected for metastases at any other site in either cluster ($q > 0.05$, Fisher’s exact test); (Supplementary Figure S1.3).

These data suggest different mutational processes may be active in PCBM compared to PCa and PCBM and non-brain metastases. The specific signatures showing altered representation in these comparisons implicate NER activity in the observed differences.”

3) p8-9 the analysis on PCBM specific mutations should be complemented with some more clinical characteristics such as pre-treatment history. It may well be that NF1 and RICTOR mutations arise as part of therapy resistance mechanisms not specific for PCBM (or as stated above as poor prognostic markers).

Response #3

We were able to obtain clinical data for 15 cases. This survey suggested that those patients were treated with standard therapy: 12/15 received androgen deprivation therapy (ADT) or orchidectomy;

2/15 Abiraterone; 1/15 Enzalutamide; 4/15 Docetaxel; 2/15 Cabazitaxel; 1/15 Radionuclide therapy and 1/15 Carboplatin. In our study, we compare them to the CRPC500 cohort of men who would have also initially received ADT or Taxane. While differences may exist in the extent and timing of the therapies received, we do not believe that they can explain the enrichment of mutations that were only rarely seen in a much large and diverse cohort (i.e., CRPC500).

Changes to manuscript #3

To acknowledge this limitation, we have modified the text in the discussion (page 18-19) to address potential limitations as follows:

“The current study does have some limitations. We identified a population enriched for PCBM. Clinical information was available for only 12 cases. These cases were treated with ADT or orchidectomy. In this study, we did not interrogate patient matched non-brain metastases. These samples were not available. Our study design addressed this deficit by comparing the PCBM with the CRPC500-cohort comprising non-brain metastases from multiple sites. The extent of tumor burden in the PCBM cohort as compared to the CRPC500 might help explain difference in the higher mutation burden seen in the metastasize to the brain. Unfortunately, the CRPC500 data does detail the extend of tumor burden. Finally, we do not propose a specific functional mechanism at this point, but hope that future work will allow us better understanding of metastatic spread to the brain.”

4) page 11 the text on the analysis of NF1 is difficult to follow. Please clarify as I cannot reconstruct the numbers...

Response #4

We apologize for the lack of clarity in this section. We have now simplified this overcomplicated description.

Changes to manuscript #4

We have modified the section on page 10 describing the prevalence of NF1 mutations to:

“Interestingly *NF1*, a gene we observed to be frequently mutated, also showed frequent SCNA, with loss in 32.1% of patients (9/28). Among the 9 patients showing *NF1* loss, seven showed loss in their metastatic samples (25%), although this was not significantly above the rate of 30.3% (126/416) observed in the CRPC500 cohort ($q > 0.05$, Fisher’s exact test). Two of the seven patients with *NF1* loss in their metastatic samples also showed *NF1* loss in their primary samples. In two other patients, *NF1* loss was detected exclusively in the primary tumor.”

5) the clonal selection analysis seems to be somewhat disconnected from the rest of the paper. Although in itself the analysis seems to have been performed in a proper manner the conclusion that the ability to metastasize to the brain is an early event is odd. With only 2 patients analyzed in this manner and no conclusive brain metastasis driver genes, this conclusion seems overstated. The analysis, in my opinion, does not seem to contribute to the overall message of the paper.

Response #5

We agree with the reviewer that this is a limitation and have modified the text to reflect that these can only be viewed as case studies. We apologize for the lack of clarity as we simply meant to state that the clones which metastasized did not arise late in the clonal evolution of the tumors.

Changes to manuscript #5

We have changed the text on page 14 to more accurately describe our findings and the limited number of cases on which clonality analysis was performed. The text on page 14 now reads:

“While clonality analyses were only performed on samples from two patients, and therefore serve primarily as case studies, these data support a model of clonal selection in the metastatic setting in these cases, and demonstrate the continued clonal evolution occurring at both the metastatic and primary sites after metastasis has occurred”.

6) Overall the conclusion that there seems to be an enrichment in the HRD deficient tumors of PCBM can be viewed as the most important conclusion. An analysis aggregating the findings: NF1/RICTOR and HRD would complement the manuscript, are they mutually exclusive or overlapping?

Response #6

We thank the reviewer for this comment. We have generated a new supplementary figure which demonstrates the overlap between HR deficient and NF1 altered patients.

New Supplementary Figure S2.5 (related to Fig. 2). NF1, RICTOR and HR-genes status across PCBM cohort. Summary of copy number and coding mutations affecting HR genes, NF1 and RICTOR in PCBM-metastases from 106 tumor samples from 28 PCBM patients. SCNA are represented by the large rectangles and small squares inside rectangles represent mutations; the effects of the SCNA and coding mutations are color-coded according to the legend. Samples are grouped by patient and sample site (primary tumors or metastasis) as indicated above the plot.

Changes to manuscript #6

We have added **Supplementary Figure S2.5** to the manuscript and added a reference to it on page 12 stating: “We detected NF1 alterations (mutation or CN loss) in patients P1, P8 and P27 which harbored biallelic inactivation of one of the PROfound HR genes.”

Reviewer #2

The manuscript by Rodriguez et al., “The Genomic Landscape of Prostate Cancer Brain Metastases” describes the genetic analysis of 28 prostate cancer cases with brain metastases (PCBM) using WES and targeted RNA sequencing, and focuses on the identification of distinct alterations enriched in the PCBM cohort by comparing to the Stand Up to Cancer/PCF CRPC500 cohort (416 prostate cancer patients with non-brain metastases). Matched primary PCa from 10 patients were also analyzed. Since brain metastasis rarely occurs in prostate cancer, it’s appreciated that this study assembled a cohort of these uncommon samples.

Here, the authors highlight frequent aberrations in *NF1* (and *RICTOR*) and signature of homologous repair defects in brain mets that they claim represent potential precision therapeutic avenues. Below are some aspects that should be addressed.

Major concerns:

1) FFPE samples are often sub-optimal and resulting NGS-based variant detections are prone to false positives. Therefore, it is curious to find that both PCBM primary tumors and brain metastases had significantly higher levels of mutations than the TCGA PRAD and the CRPC500 cohorts, when these two publically available cohorts are composed of a high ratio (if not all) of frozen tissues. It's not obvious from the manuscript that if the following factors have been considered.

(1.1) Enzymatic repair of FFPE DNA can help to reduce false mutation frequencies by removing artefactual C-T conversions that result from cytosine deamination during formalin fixation. Was the step included in the sample preparation process?

(1.2) The authors set a threshold to filter out those SNVs and indels with a variant allelic fraction (VAF) of <1% and/or those supported by <3 reads. Can the settings efficiently eliminate potential FFPE artifacts that are prone to low allele frequency? False discovery assessments according to variant allele frequency should be implemented.

Response #1.1, 1.2

We agree that FFPE artefacts can significantly affect interpretation of these data, and have taken appropriate steps to mitigate these. We have included a table summarizing all mutations which includes the VAF for each variant (Supplementary data SD5). While we did not perform enzymatic repair of the DNA, analysis of discordance in called variants between FFPE replicates demonstrated that the majority of FFPE artefacts have a VAF in the range 1-4% (Bhagwate et al, PMID: 31477010). Furthermore, comparison of the post-filtering VAFs of C>T/G>A SNVs with those of other SNVs in this study showed no difference in the mean VAF of these variants compared to other SNVs.

New SD5. Summary mutation call for PCBM-cohort.

Changes to manuscript #1.1, 1.2

We have added details describing the filtering of variants in the text and included a new data sheet to the Supplementary Data (**SD5**) with the VAF of all mutations so that the efficacy of our filtering can be assessed. The text on page 23 now reads:

"We further excluded variants identified in at least two of a panel of 123 non-tumor samples, including the non-tumor samples included in the current study, captured and sequenced using the same protocols using the artifact detection mode of MuTect2 implemented in GATK, which includes steps to remove Oxo-G and FFPE artefacts as previously published⁵¹".

1.3) A complete list of mutation calls with detailed information and VAF should be provided.

Response #1.3 Thank you for pointing out this omission. We have added this information for all mutations in a new supplementary data table (SD5).

Changes to manuscript #1.3

We have added a new data sheet to the **supplementary data (SD5)** that includes the detailed information for all mutations.

1.4) Were the TCGA and CRPC500 cohort reanalyzed by the same pipeline? Data should be treated the same way for direct comparison.

Response #1.4

Numerous high impact studies based on WES have similarly compared mutation calls produced in-house, to TCGA datasets e.g. Bertucci et al (PMID 31118521), Witkiewicz et al. (PMID: 25855536), Wedge et al. (PMID: 29662167). While this is a limitation in studies comparing mutation calls arising from different pipelines/callers, we feel the enormous amount of data which must be obtained and processed in order to do this means we cannot reasonably complete this request.

Changes to manuscript #1.4

No changes were made to the manuscript regarding this point.

1.5) Are matched normal samples sequenced for each patient? Tumor-alone sequencing could add another level of noise in mutational analysis.

Response #1.5

Matched normals were used for 19/28 samples (as shown in Supplementary data SD1). For those without matched normals, we used a pool of normals consisting of the 19 normals from this study, plus an additional 22 normal prostate samples from a separate study. We further filtered to remove variants in at least two of a panel of 123 non-tumor samples.

Changes to manuscript #1.5

We have extended the methods section to describe the use of matched normals with better clarity. The text on page 23 now reads:

“Matched normals were used as reference for 19/28 samples (**Supplementary data SD1**). For those without matched normals we used a pool of normals consisting of the 19 normals from this study, plus an additional 22 normal prostate samples from a separate study”.

2). Clinical significance of brain-mets as a unit of analysis is not clear. Do prostate cancers that are positive for brain-mets show a different metastatic profile for other sites? Do these show more aggressive clinical course than cases with non-brain-mets, different survival, different histomorphology, or different immune infiltration profile?

Response #2

The presence of brain metastases is generally a poor prognostic factor for patients with cancer. Patients with brain metastases are mostly excluded from clinical trials because of their dismal prognosis.

In our view, there is insufficient published evidence to conclude that these tumors exhibit a different clinical pattern. In historic series it appears to be a late event associated with disseminated disease and high tumor burden. In a series from the M. D. Anderson Cancer Center published in 2003 (Tremont-Lukats et al., PMID: 12872358), the authors identified 103 cases with parenchymal brain metastases from a cohort of 16,280 patients with prostate cancer. In this cohort, median time to BM detection was 35 months from diagnosis of prostate cancer, with following median survival of another 1 month afterwards. Concomitant bone metastasis were reported in 95% of patients; lung in 31%, and liver in 19%. There was no direct comparison with the survival data of patients without BM.

We believe that prostate cancer with metastases to the brain constitute a small but relevant subgroup (up to 2%) of a highly prevalent disease. Analogous to other common cancers, the continuous improvement of systemic treatment options for metastatic prostate cancer patients may lead to higher incidence of BM in the next years.

We obtained clinical information from 15/28 patients. In addition to the brain metastases, these 15 patients harbored metastases in bone and/or lymph nodes. Moreover, 6/15 patients also presented liver and/or lung metastases. A comparison of these data is not possible with CRPC500-cohort, since similar information is not available for the CRPC500 (please see also response #5).

The pathology of the metastases in the PCBM cohort was found very similar to the CRPC500-cohort including adenocarcinomas (89.3% vs 88.8%) and prostate cancer with neuroendocrine features (10.7% vs 11.2%), respectively. No formal immune landscape analysis was performed but pathology review of both cohorts revealed a minimal level of lymphocytic infiltration (MAR reviewed all cases from both cohorts). The pathology of both metastatic cohorts shown strong similarities including adenocarcinomas (89.3% vs 88.8%) and prostate cancer with neuroendocrine features (10.7% vs 11.2%), respectively.

In summary, we feel that probably more than differences in clinical pattern of disease, patients with BM from prostate cancer exhibit biological particularities, which could be therapeutically exploited.

Changes to manuscript #2

We described in the manuscript (page 5) the similarities in the pathology of PCBM metastases with the non-brain metastases in the CRPC500-cohort. We also have acknowledged this limitation in the discussion (page 18-19). The text now reads:

Page 5

“This is a similar distribution of morphologic phenotypes as compared to the recent studies by Abida et al. where 89% cases were classified as adenocarcinoma and 11% of the CRPC cases manifested NE features¹⁰.”

Page 18-19

“The current study does have some limitations. We identified a population enriched for PCBM. Clinical information was available for only 12 cases. These cases were treated with ADT or orchidectomy. In this study, we did not interrogate patient matched non-brain metastases. These samples were not available. Our study design addressed this deficit by comparing the PCBM with the CRPC500-cohort comprising non-brain metastases from multiple sites. The extent of tumor burden in the PCBM cohort as compared to the CRPC500 might help explain difference in the higher mutation burden seen in the metastasize to the brain. Unfortunately, the CRPC500 data does detail the extend of tumor burden. Finally, we do not propose a specific functional mechanism at this point, but hope that future work will allow us better understanding of metastatic spread to the brain.”

3). *The cohort size of brain-mets analyzed here, 28 metastases/ 10 matched PCa- is higher than any previous studies- but then no previous study purported to describe brain-met specific genomic aberrations. Here the cohort size prompts concern for incidental over-representation of some aberrations as brain-met specific association. The authors need to support their NF1/ RAS pathway association with functional analysis, since data from prostate brain-mets is fairly scant in literature.*

Response #3

This study was designed to define the genomic landscape of PCBM as they have only rarely been studied. We used the CRPC500 cohort as a large historical comparator. The enrichment observed is informative as it nominates NF1/RAS but also some other genes as being preferentially enriched. We believe that these findings have generated hypotheses that will require detailed functional studies including mouse model xenograft studies to determine the effect of these individual mutations on the probability of developing or maintaining brain metastases. We have now added text in the discussion

outlining the next steps needed to credential these mutations. However, in the current study we view this functional work as out of the scope of work.

Changes to manuscript #3

We have mentioned in the discussion that future work can address functional questions raised in this study. The text on page 19 now reads:

“...Finally, we do not propose a specific functional mechanism at this point, but hope that future work will allow us better understanding of metastatic spread to the brain.”

4). A comparator missing from the study is patient matched non-brain mets. This would have helped compare/contrast if tumor cell clones establishing as brain-mets are qualitatively distinct from non-brain-mets. Considering that the authors have access to brain-mets from rapid autopsy program- wherever brain-mets are obtained from, those cases do certainly have metastatic tissues in other sites as well, in much higher abundance if anything. A direct comparison of brain- and non-brain-mets from the same cases could help directly identify any brain-met specific aberrations.

Response #4

The reviewer makes an important point. Unfortunately, we are only partially able to address it. Most of the brain metastasis reported in the autopsy series were dural lesions, and in Switzerland we do not have access to additional Rapid Autopsy material to address these questions. Our study design, albeit less than perfect, was intended to address this question by comparing non-PCBM from multiple sites. We have addressed this as an additional limitation to the study.

Changes to manuscript #4

To address this limitation, we have modified the text in the discussion (page 18-19), as described in response#2.

5). The authors need to distinguish the possibility that in some cases, prostate-mets tend to preferentially metastasize to brain, rather than other organ sites; from the other, simpler possibility that presence of brain-mets are merely indicative of a higher metastasis burden: more mets, go to more places. If the former possibility be supported by data, analysis of brain-met specific aberrations would be moot. However, as seen in Figure 1 and elsewhere, arguably, mCRPCs that are seen to display a higher mutation burden, are presumably associated with a higher burden of metastasis sites- including as far as brain. The authors need to rule out this possibility to rationalize a brain-met specific hypothesis.

Response #5

The extent of tumor burden is an important point. As suggested by the Reviewer, we posit that the higher mutation burden is characteristic of tumors that metastasize to the brain. However, we do not propose a specific mechanism at this point. We queried the SU2C CRPC500 clinical leader, Dr. Wassim Abida (Personal communication, June 18th, 2020), as to the availability of tumor burden data. He responded that there is no annotation of tumor burden in the SU2C project. There was only annotation of the site that was biopsied for the sequencing. Therefore, we can only propose to highlight the differences with regards to the interrogated tumor samples as has been done in this study. (Please see also response #2; 4th paragraph).

Changes to manuscript #5

We have now noted the lack of tumor burden information as a possible confounder in the discussion sections (page 18-19), as described in response 2.

6. The possibility that PCBMs are simply associated with higher “mets-burden”, makes comparison of PCBMs with non-brain mets across public data less attractive- as many of those cases may simply represent less-disseminated mets cohorts.

Response #6

Please see response to question #5.

Changes to manuscript #6

Please see response to question #5.

7. The 28 PCBMs cases included here- what is the burden of non-brain mets in this case. Do these cases show any distinct pattern of metastatic sites compared to cases without PCBMs?

Response #7

Please see response #2; 4th paragraph.

Changes to manuscript #7

Please see response #2.

8). Please show mutation burden (Figure 1, S1.2, etc.) in the more standard format of “number of mutations/ Mb sequencing data”. This is important to normalize the data across the dataset, subject to variations in coverage, as well as outside data, in public domain/ publications. Figure 1b, the y-axis scale of the one extreme outlier PCA sample can be broken, to help scale the rest of the graph more appropriately.

Response #8

Thank you for this suggestion. We have changed this in Figure 1a, 1b and S1.2 and where appropriate in the text. We have also included a table summarizing the tumor mutational burden (TMB) for SNVs and indels (Supplementary data SD4, see details in response#1 to Reviewer1).

Changes to manuscript #8

We have changed all mutational counts to TMB and summarized these in a new data sheet in **Supplementary Data (SD4)**.

We have changed **Figure 1a. and 1b, and S1.2** to depict TMB rather than mutation count and the text describing this.

The text on page 5-6 now reads:

“The 106 samples (39 from primary tumors and 67 from metastases) from 28 patients used for this study were sequenced to a median depth of coverage of 243x for primary and of 210x for metastases. Somatic mutation analysis (see methods) identified a total number of non-synonymous mutations (SNVs and InDels) ranging from 0.64 – 50.98 mutations/Mb per sample (median 6.06 somatic mutations/Mb) (Figure 1a). The PCBMs primary samples had a median of 5.15 somatic mutations per Mb per sample (range 2.60 - 50.98), while PCBMs metastases samples had a median of 7.25 somatic mutations per Mb per sample (range 0.64 - 46.75). There was no significant difference in the number of somatic mutations, SNVs alone (median 4.79 primary, 6.10 metastases), deletions (median 0.34 primary, 0.34 metastases) or insertions (median 0.17 primary, 0.20 metastases) observed between primary and metastatic samples ($q > 0.05$, Wilcoxon test) (Supplementary Figure S1.2). Across all samples, the highest mutation rate was consistently less than 23 mutations/Mb per sample, with the exception of patient P1, whose disease harboured an MLH1 somatic missense mutation

(p.Arg522Trp) coupled with the copy number loss of the other allele, with mutation rates ranging from 27.00 to 50.98 mutations/Mb per sample (**Figure 1a**). However, we ran microsatellite instability (MSI)-sensor and additionally analyzed the MSI-status through the clinically applied Bethesda panel (AppliedBiosystems) and could not detect any evidence of MSI in any tumor sample from patient P1 (data not shown).

Metastatic samples from the PCBM cohort had significantly higher levels of non-synonymous single nucleotide variants (SNV), insertions and deletions than non-PCBM metastatic PCa samples ($q < 0.01$, Wilcoxon test). Primary samples from the PCBM cohort also had significantly higher levels of SNVs, insertions and deletions than the TCGA primary cohort. ($q < 0.01$, Wilcoxon test) (**Supplementary Data SD4**).

The methods section was also updated accordingly in page 24 as follows:

“Tumor mutational burden (TMB) for each mutation type for each sample were calculated using the exome size for the Agilent Human All Exon V7 panel for samples sequenced in this study, the Agilent SureSelect Human All Exon V4 panel for the CRPC500 cohort, and 38 Mb for the TCGA data⁵⁵.”

9). *“The mismatch repair-defective PCA (P1), harboring an MLH1 somatic missense mutation (p.Arg522Trp) coupled with the copy number loss of the other allele, with mutation rates ranging from 964 to 1820 mutations/sample (Figure 1a)”. Can the authors confirm if this sample shows microsatellite instability (MSI) signature?? Incidentally, p.Arg522Trp is annotated in ClinVar as a germline variant of unknown significance; in COSMIC this mutation is noted only in one case of CRC with unknown status of MSI. The mutational signatures shown in Figure 1a doesn't seem to be consistent with MMRD for this patient, either.*

Response #9

We thank the reviewer for pointing this out. The reviewer was correct in that the MLH1 mutation was a rare variant with no evidence of malignant potential. Concordantly, we applied MSIsensor and analysed the MSI-status of all samples from patient 1 through the clinical applied Bethesda panel (AppliedBiosystems) and could not detect any evidence of MSI.

Changes to manuscript #9

We included this analysis in the manuscript. The text on page 6 now reads:

“However, we ran microsatellite instability (MSI)-sensor and additionally analyzed the MSI-status through the clinically applied Bethesda panel (AppliedBiosystems) and could not detect any evidence of MSI in any tumor sample of patient P1 (data not shown).”

10). *The mutations included in Figure 2 and 3: please provide a detailed list of specific mutations with nucleotide/ amino-acid changes showing specific aberrations, along with concomitant status of copy number or copy neutral LOH.*

Response #10

Thank you for highlighting that this was missing, we apologize for not having included this in the previous version. We have added this information in the supplementary data.

Changes to manuscript #10

We have added a table including these details for all mutations (**Supplementary data SD5**), and copy number data in a separate table (**Supplementary data SD8**).

11.1) SCNA data- Figure S2.3- Can this be parsed to highlight SCNAs enriched in the PCBM?

Response #11/1

We apologize for the lack of clarity. We have further highlighted genes enriched for SCNAs in the PCBM on the bottom track of Supplementary Figure S2.3.

Changes to manuscript #11.1

We have changed to green the labelling of genes enriched for SCNAs in PCBM in Supplementary Figure S2.3.

11.2) *Figure S2.2 a and b figure labels are identical- need to fix.*

Response #11.2

Thanks for pointing that out, we have corrected this.

Changes to manuscript #11.2

Label of axis of Figure S2.2 b now reads "Difference in relative frequency of alteration CNA + mutations".

12). *In this study, it seems that all mutations are viewed without the assessment of "driver vs. passenger" or "biological significant vs. variant of unknown significance". A good example is RICTOR. Mutations of this gene were reported to be significantly enriched in the PCBM cohort. Hence, possible activation of the druggable pathway PI3K/AKT/mTOR was indicated. However, RICTOR is not an oncogene with well-known hotspot activating mutations. In addition, positions of RICTOR mutations were not actually provided in the manuscript for evaluation. More evidence needs to be provided to call clinical relevance of RICTOR mutations; ie: Page 11: "observed mutations affecting RICTOR (100% missense)"- please provide the details of the mutations to allow an evaluation if these are likely functionally relevant or not.*

Response #12

Thank you for this important point. We have now included this information in Supplementary data SD5.

Changes to manuscript #12

We have added **supplementary data SD5** that includes this information.

13). *In Materials and Methods and elsewhere- no indication of assessment of LOH by copy neutral allelic imbalance was noted. Related to that, this study does not distinguish monoallelic- and biallelic-inactivation of tumor suppressor genes. Assessments of tumor suppressors (such as NF1, PTEN, TP53, RB1, BRCA1) based on "one-hit" events (heterozygous loss or mutation without second hit) are not meaningful for this class of genes.*

Response #13

Thank you for highlighting this. We have included LOH with the summary of mutations (SD5) along with a table of CNAs (SD8) so that biallelic inactivation can be more easily assessed.

New SD8. Summary somatic copy number alterations SCNA for PCBM-cohort**Changes to manuscript #13**

We have included LOH with the summary of mutations (**Supplementary Data SD5**) along with a table of CNAs (**Supplementary Data SD8**).

14). *In the highlighted PROFOUND clinical trial (de Bono, J. et al. 2020, Olaparib for Metastatic Castration-Resistant Prostate Cancer. NEJM 382, 2091-2102), a key observation seems to be the rather insignificant responsiveness of non-BRCA HR mutants to PARP inhibitor therapy. It doesn't really support the inclusion of the wider net of HR gene aberrations as proposed here- please comment/ revise the text.*

Response #14

We appreciate the Reviewer's point that our study should just highlight the Group A genes (i.e., BRCA1/2 and ATM). However, as this study is meant to fuel new avenues of research to understand potential mechanisms of drug resistance, we wanted to provide a broader range of genes that were considered. For example, our laboratory has previously demonstrated that FANCA mutations may confer sensitivity to PARPi (Wilkes et al., PMID [28864460](https://pubmed.ncbi.nlm.nih.gov/28864460/)). The reason the *PROFOUND* study does not show response may be due to small numbers but not biology. Therefore, to ensure that researchers interested in homologous repair mechanisms have access to a comprehensive dataset, we feel it would be important to maintain the border list. We have added to the discussion that only BRCA1/2 and ATM mutations were approved by the FDA for therapy with PARPi.

Changes to manuscript #14

We added to the discussion the approved criteria by FDA for receiving PARPi (Olaparib) and pointed out the major benefits obtained in patients harboring BRCA1/2 and ATM alterations. The text on pages 17- 18 now reads:

"As suggested by the results from the PROfound and TOPARP-B trials^{22,34,35}, PARP inhibition(Olaparib) is an effective therapy and was approved by FDA in May 2020 (<https://www.fda.gov/drugs/drug-approvals-and-databases/fda-approves-olaparib-hrr-gene-mutated-metastatic-castration-resistant-prostate-cancer>) in patients with metastatic prostate cancer after relapse to enzalutamide and abiraterone and harboring deleterious or suspected deleterious germline or somatic homologous recombination repair (HRR) gene-mutated."

15). *The proposed equivalence of NF1 in PCBM with YAP1 mutations in lung cancer is highly speculative- YAP1 is well characterized as a regulator of Hippo pathway. The one cited cross-talk with RAS pathway is likely context specific, not its central signaling axis.*

Response #15

This is a good point. We have revised the discussion.

Changes to manuscript #15

The text on page 17 now reads:

"Shin et al. also provide *in vivo* data supporting the functional relevance of YAP1 in enabling lung cancer brain metastases. Although context specific, there is cross talk within the RAS-pathway raising the question of similarities between the two cancer types. Further investigation will be needed in order to validate this interaction."

Minor Points:

1). *In Supplementary Figure 1, 4/24 should be 4/28.*

Thank you for this note, we have corrected this typo in the **Supplementary Figure 1**.

2). *“PIK3CD” is misspelt in a few places as “PI3KCD”.*

Thank you for this note, we have corrected the text.

Reviewers' Comments:

Reviewer #1:

Remarks to the Author:

the authors have very thoroughly addresses the comments and I have no further comments

Reviewer #2:

Remarks to the Author:

The revised manuscript provides many clarifications and additional analyses to support the main conclusions of the manuscript. However, the primary concern remains essentially unresolved: whether "prostate brain mets" (PCBMs) represent a distinct molecular entity or simply outlier cases with higher burden of metastasis. Also the question of comparative analysis of mutation burdens seems not fully resolved. Overall, more questions are being raised than answered and the manuscript is not suitable for publication in its current form. Specific issues still outstanding are:

1. Is the difference in mutation burden between TCGA and CRPC500 statistically significant as has been noted previously (PMID: 31824860, PMID: 28783718)? Visually it doesn't appear so. In comparison, the mutation burden of PCBM primary and mets seems much too similar (and higher than external cohorts). The offered explanation in terms of different "tumor burden" seems unsatisfactory- as primary tumor always tends to display lower tumor content and mutation burden than mets- but here, both seem similar in PCBM cohort and dramatically different from TCGA and CRPC500.
2. Bhagwate et al. demonstrated that the majority of FFPE artefacts have a VAF in the range 1-4%. With that, isn't it more reasonable to set the cutoff at 5% VAF to eliminate FFPE noise, especially when tested samples of this cohort have decent tumor content? The authors need demonstrate that a threshold as low as 1% VAF and 3 supporting reads is valid before comparing the results to the TCGA and the CRPC500 cohorts.
3. The new supplementary data table SD5, Summary mutation call for PCBM-cohort: among total 2,466 clonal mutations (by pyclone), 2,331 are labelled pass filter; however, of the 20,013 subclonal mutations, only 7,441 are passed filter. Facets clonal analysis has comparable numbers. Authors, please confirm if all mutations or only passed filter ones are used for estimation of mutation burden and comparison across cohorts? Is the large number of subclonal mutations that do not pass filter accounting for an inflated mutation burden in PCBM samples (due to FFPE artifacts)?
4. Also, in new supplementary data table SD5: The authors write in the Methods "We filtered out SNVs and indels outside of the target regions, those with a variant allelic fraction (VAF) of <1% and/or those supported by <3 reads." However, a large fraction of mutations (>7,000 mutations) in SD5 showing VAF as low as 0.1% or 1 supporting read are included in the mutation list (Column O, Tumor FA; Column S, Tumor AD). This table appears not to be properly filtered, but the results are presented in Fig 1a.
5. The mutation burden in TCGA pegged at 0.55 in Supplementary Table SD4, seems nearly half of this value in a previous reference formally analyzing mutation burden in different prostate cancer cohort studies (PMID: 31824860). Even taking the numbers at face value- mutation burden in TCGA versus CRPC500- the difference is close to 2 fold. However, the difference in mutation burden between primary and mets in PCBM cohort is marginal in comparison. This merits an examination as well to ensure consistency and accuracy.
6. The noted mutations in NF1 (6 samples) highlighted in the first draft: looking at the specifics of the mutations, NF1, only eight out of 24 mutations (passed filter) were found to be deleterious (stopgain/ frameshift)- with only 3 of these showing LOH (Table SD5). Additionally, it's surprising to find that near all NF1 mutations are subclonal with low VAF (many of them are <1% and <3 reads). The only exception - patient 33 (19-33-5-2-33-M-S17) harbors two missense mutations (L1064I and E1436G) with unknown significance. First of all, the mutation list needs to be properly cleaned-up. Secondly, drivers of brain mets are expected to be clonal events in the brain tumor samples, but it's not the case in this study. It is not convincing that NF1 mutations are enriched in PCBM, or NF1 contributes to brain metastasis.

7. RICTOR (5 samples) highlighted in the first draft: no recurrent mutations/ involving any of the conserved amino acids were noted in RICTOR- which incidentally in literature have been shown to be more commonly amplified in subsets of cancers, representing gain of function by overexpression (PMID: 29955840). The 5 RICTOR mutations reported here are R336H, E1192A, D1478G, A713G and D1414H. Although all clonal events, none of these variants has been functionally characterized. Evidence is needed, either by literature support or functional studies, to claim pathogenic roles of these mutations.

8. On page 6 the authors write "The signature for defective homologous repair (SBS3) was observed in the metastases of two patients (P9 and P30)..". What are the genetic causes of this phenotype in P9 and P30? Similarly, on page 11 the authors write "Patient P1 harbored BRCA2 nonsense mutations in both their metastatic and primary samples. Patient P23 showed a BRCA2 frameshift mutation and patient P36 a missense mutation within their metastatic samples". Why in Figure 1, the defective homologous repair (SBS3) signature not observed in patients P1, P23 or P36 if those events are pathogenic? The authors should make an effort to associate genotypes and phenotypes. Otherwise it's difficult to follow the points this study is trying to make.

9. There is patient 37 in Supplementary data SD5, but no patient 36. Patient 37 is not mentioned in any of the figures.

10. Supplementary Figure S1.2 (related to Fig. 1): Replace SNPs with SNVs? SNPs typically refer to germline variants.

11. Supplementary Figure S2.2 (related to Fig. 2): change TGCA to TCGA

12. Multiple typos in this one sentence: "The extent of tumor burden in the PCBM cohort as compared to the CRPC500 might help explain difference in the higher mutation burden seen in the metastasize to the brain. Unfortunately, the CRPC500 data does detail the extend of tumor burden."

13. Figure 1b, the y-axis scale of the one extreme outlier PCA sample can be broken to help scale the rest of the graph more appropriately.

Reviewer #3:

None

NCOMMS-20-18280A: Point by Point Response to Reviewers' Comments

Reviewer #1

The authors have very thoroughly addressed the comments and I have no further comments

Response: We thank the Reviewer for the supportive comment.

Reviewer #2

The revised manuscript provides many clarifications and additional analyses to support the main conclusions of the manuscript. However, the primary concern remains essentially unresolved: whether “prostate brain mets” (PCBM) represent a distinct molecular entity or simply outlier cases with higher burden of metastasis. Also the question of comparative analysis of mutation burdens seems not fully resolved. Overall, more questions are being raised than answered and the manuscript is not suitable for publication in its current form. Specific issues still outstanding are:

1. Is the difference in mutation burden between TCGA and CRPC500 statistically significant as has been noted previously (PMID: 31824860, PMID: 28783718)? Visually it doesn't appear so. In comparison, mutation burden of PCBM primary and mets seems much too similar (and higher than external cohorts). The offered explanation in terms of different “tumor burden” seems unsatisfactory- as primary tumor always tends to display lower tumor content and mutation burden than mets- but here, both seem similar in PCBM cohort and dramatically different from TCGA and CRPC500.

Response #1 There were some methodological differences regarding the computation of the mutation burden between TCGA, CRPC500 and the PCBM cohorts. To address these concerns, we unified the approach to ensure they are as comparable as possible. We have amended the method used to calculate TMB in line with Ryan and Bose (2019, PMID: 31824860). We detected significantly higher levels of alterations (SNVs, insertions and deletions) in the PCBM metastatic samples compared to the CRPC500, and the PCBM primary samples compared to the TCGA cohort ($q < 0.001$, Wilcoxon test). Significantly more SNVs, insertions and deletions were present in the CRPC500 samples compared to the TCGA ($q < 0.001$, $q < 0.001$ and $q < 0.01$ respectively Wilcoxon test). Indeed, we detected no significant difference in the number of SNVs, insertions or deletions between PCBM primary and metastatic samples ($q > 0.05$, Wilcoxon test). (See also response to comment #5).

New Figure 1b

New Supplementary Figure S1.2

Changes to manuscript #1

We have changed Figure 1b, supplementary Figure S1.2 and SD4 to depict these new values.

2. Bhagwate et al. demonstrated that the majority of FFPE artefacts have a VAF in the range 1-4%. With that, isn't it more reasonable to set the cutoff at 5% VAF to eliminate FFPE noise, especially when tested samples of this cohort have decent tumor content? The authors need demonstrate that a threshold as low as 1% VAF and 3 supporting reads is valid before comparing the results to the TCGA and the CRPC500 cohorts.

Response #2

We appreciate this comment. We examined the proportion of C>T and G>A transition mutations in the low VAF (<5%) and high VAF (>5%) variants and, as shown in the below figure, found no enrichment for these mutations in the low VAF variants. As the reviewer rightly notes, most FFPE artefacts are found at a VAF <5%. We believe that this demonstrates the mutations called by our pipeline are highly unlikely to be influenced by FFPE artefacts.

Changes to manuscript #2

None.

3. The new supplementary data table SD5, Summary mutation call for PCBM-cohort: among total 2,466 clonal mutations (by pyclone), 2,331 are labelled pass filter; however, of the 20,013 subclonal mutations, only 7,441 are passed filter. Facets clonal analysis has comparable numbers. Authors, please confirm if all mutations or only passed filter ones are used for estimation of mutation burden and comparison across cohorts? Is the large number of subclonal mutations that do not pass filter accounting for an inflated mutation burden in PCBM samples (due to FFPE artifacts)?

Response #3

We apologize for the lack of clarity in the presentation of SD5. As in our previously published studies, e.g. Ng et al (2017, PMID: 28351929) and Shi et al. (2018, PMID: 30404001), in our analysis, we first called the somatic mutations in each tumor sample, then we genotyped the positions of all somatic mutations in all tumors from the same patient to retrieve variants that may be present below the detection limits of the variant callers but were nonetheless present (described in the methods, but now clarified further). This second step is to avoid false negative calls and to aid clonality analysis. Excluding these mutations does not change the finding of increased TMB in our cohort compared to TCGA and CRPC500. The non-

pass variants the reviewer referred to are those identified by genotyping in the second step. These were labeled in SD5 but their labeling was not clearly described in the methods. We have now clarified the labelling of these variants.

Changes to manuscript #3

The methods on page 25 now reads:

Variants called through genotyping are annotated as 'Allele specific genotyping' in SD5.

4. Also, in new supplementary data table SD5: The authors write in the Methods "We filtered out SNVs and indels outside of the target regions, those with a variant allelic fraction (VAF) of <1% and/or those supported by <3 reads." However, a large fraction of mutations (>7,000 mutations) in SD5 showing VAF as low as 0.1% or 1 supporting read are included in the mutation list (Column O, Tumor FA; Column S, Tumor AD). This table appears not to be properly filtered, but the results are presented in Fig 1a.

Response #4

As with the previous comment, this was unclear due to a lack of clarity regarding the values in the 'Filter' column in SD5. We have now added an explanation of this in the methods.

Changes to manuscript #4

Please see response #3.

5. The mutation burden in TCGA pegged at 0.55 in Supplementary Table SD4, seems nearly half of this value in a previous reference formally analyzing mutation burden in different prostate cancer cohort studies (PMID: 31824860). Even taking the numbers at face value- mutation burden in TCGA versus CRPC500- the difference is close to 2 fold. However, the difference in mutation burden between primary and mets in PCBM cohort is marginal in comparison. This merits an examination as well to ensure consistency and accuracy.

Response #5

We have revised the computation of the mutational burden as described in the response #1.

We have recalculated the TMB estimates for the PCBM samples, as well as the TCGA and CRPC500:

- The revised TMB for the TCGA samples is 0.945, reflecting the TMB reported in Ryan and Bose (2019, PMID: 31824860) for the same cohort (0.94 NS SNVs/Mb) (see updated Figure 1b, supplementary Figure S1.2 and SD4).
- The revised TMB for the CRPC500 cohort is 1.57; slightly lower than the values reported for metastatic samples in Ryan and Bose (2019, PMID: 31824860) which range from ~2 - ~4 NS SNVs/MB (see updated Figure 1b, supplementary Figure S1.2 and SD4).
- There was no significant difference in the number of SNVs, insertions or deletions detected in PCBM metastasis compared to PCBM primary samples ($q > 0.05$, Wilcoxon test) (see updated Figure 1b, supplementary Figure S1.2 and SD4).

These data reflect those observed in the clonality analysis, whereby we observed the continued acquisition of mutations, and clonal evolution, in the primary samples even after metastasis had occurred. In conjunction with the high-grade morphology of the primary samples, the wide range in the number of mutations detected (0.64 – 50.98/Mb) and the

low number of samples, we believe that this may explain the high TMB observed in the primary samples, and the lack of significant difference in TMB between primary and metastatic samples.

Changes to manuscript #5

We have changed Figure 1b, supplementary Figure S1.2 and SD4 to depict these new values.

6. The noted mutations in NF1 (6 samples) highlighted in the first draft: looking at the specifics of the mutations, NF1, only eight out of 24 mutations (passed filter) were found to be deleterious (stopgain/ frameshift)- with only 3 of these showing LOH (Table SD5). Additionally, it's surprising to find that near all NF1 mutations are subclonal with low VAF (many of them are <1% and <3 reads). The only exception - patient 33 (19-33-5-2-33-M-S17) harbors two missense mutations (L1064I and E1436G) with unknown significance. First of all, the mutation list needs to be properly cleaned-up. Secondly, drivers of brain mets are expected to be clonal events in the brain tumor samples, but it's not the case in this study. It is not convincing that NF1 mutations are enriched in PCBM, or NF1 contributes to brain metastasis.

Response #6

We thank the reviewer for this point, as we agree that the low CCF of *NF1* mutations is an important factor to consider when assessing these findings. We have clarified the inclusion of these mutations in SD5, as per the responses to comments #3 and #4.

Changes to manuscript #6

We have changed the text on page 12 to reflect this. The text now reads:

It is important to note mutations in NF1 were sub-clonal and the functional consequences of the detected mutations in both NF1 and RICTOR remain to be validated. However, these data highlight the overall frequency of NF1 alterations, its potential for biallelic inactivation and, in combination with the observed mutations affecting RICTOR (clonal in 4/5 patients with RICTOR alterations), suggest the possibility for downstream activation of two druggable pathways, RAS/RAF/MEK1-2/ERK1-2 and PI3K/AKT/mTOR1/2, respectively.

7. RICTOR (5 samples) highlighted in the first draft: no recurrent mutations/ involving any of the conserved amino acids were noted in RICTOR- which incidentally in literature have been shown to be more commonly amplified in subsets of cancers, representing gain of function by overexpression (PMID: 29955840). The 5 RICTOR mutations reported here are R336H, E1192A, D1478G, A713G and D1414H. Although all clonal events, none of these variants has been functionally characterized. Evidence is needed, either by literature support or functional studies, to claim pathogenic roles of these mutations.

Response #7

We do not currently have functional support for the role of the *RICTOR* although these were clonal variants. We accept that these remain to be characterized so have modified the text to reflect this.

Changes to manuscript #7

The text on page 12 now reads:

It is important to note mutations in NF1 were sub-clonal and the functional consequences of the detected mutations in both NF1 and RICTOR remain to be validated. However, these

data highlight the overall frequency of NF1 alterations, its potential for biallelic inactivation and, in combination with the observed mutations affecting RICTOR (clonal in 4/5 patients with RICTOR alterations), suggest the possibility for downstream activation of two druggable pathways, RAS/RAF/MEK1-2/ERK1-2 and PI3K/AKT/mTOR1/2, respectively.

8. On page 6 the authors write “The signature for defective homologous repair (SBS3) was observed in the metastases of two patients (P9 and P30)..”. What are the genetic causes of this phenotype in P9 and P30? Similarly, on page 11 the authors write “Patient P1 harbored BRCA2 nonsense mutations in both their metastatic and primary samples. Patient P23 showed a BRCA2 frameshift mutation and patient P36 a missense mutation within their metastatic samples”. Why in Figure 1, the defective homologous repair (SBS3) signature not observed in patients P1, P23 or P36 if those events are pathogenic? The authors should make an effort to associate genotypes and phenotypes. Otherwise it’s difficult to follow the points this study is trying to make.

Response #8

We agree that the text could be improved. We have amended the text discussing this on page 6 and page 12, adding links between genotype and the observed mutational signatures.

Changes to manuscript #8

The text on page 6-7 now reads:

Substantial differences in mutational signatures were detected between primary and metastatic samples. The signature for defective homologous repair (SBS3) was observed in the metastases of two patients (P9 and P30), while it was absent in the matched primary samples from these patients⁶ (Figure 1a). Patient P9 was found to harbor a missense mutation in BRCA1, predicted to be disease-causing by MutationTaster and associated with LOH in one metastatic sample, while patient P30 showed no mutation in any of the genes included in the PROfound clinical trial¹⁸. Both patients were classified as large-scale state transition (LST) high (see Methods). The absence of an obvious lesion affecting the HR pathway, despite the presence of the HRD mutational signature does, however, reflect a recent study which reported detection of the HRD signature in a subset of patients without BRCA1/2 mutations⁹. Furthermore, BRCA1 is frequently epigenetically silenced in cancer, representing another mechanism through which HRD could arise²⁰⁻²².

The text on page 12 now reads:

Patient P1 harbored BRCA2 nonsense mutations in both their metastatic and primary samples, predicted to be disease-causing by MutationTaster, although the signature for HRD was only detected in one primary and one metastatic sample. Patient P23 showed a BRCA2 frameshift mutation and patient P36 a missense mutation within their metastatic samples. These BRCA2 alterations were associated with LOH in all three patients. While the signature for HRD was not detected in patient 23 or P36, all of these patients were classified as LST high. Therefore, while there is some evidence for HRD in samples carrying HR mutations, the pathogenicity of these mutations would require further investigation.

9. There is patient 37 in Supplementary data SD5, but no patient 36. Patient 37 is not mentioned in any of the figures.

Response #9

We have checked and can confirm the correct patient numbers are used in all figures and tables.

Changes to manuscript #9

None.

10. Supplementary Figure S1.2 (related to Fig. 1): Replace SNPs with SNVs? SNPs typically refer to germline variants.

Response #10

Thank you for pointing this out, it has been corrected.

Changes to manuscript #10

We have corrected this in the new version of the figure (see response #1).

11. Supplementary Figure S2.2 (related to Fig. 2): change TGCA to TCGA

Response #11

Thank you for pointing this typo.

Changes to manuscript #11

We have corrected the label in the figure.

12. Multiple typos in this one sentence: "The extent of tumor burden in the PCBM cohort as compared to the CRPC500 might help explain difference in the higher mutation burden seen in the metastasize to the brain. Unfortunately, the CRPC500 data does detail the extend of tumor burden."

Response #12

Thank you for this editorial point, we have improved the wording of this section.

Changes to manuscript #10

The text on page 19-20 now reads:

The current study does have some limitations. Clinical information was available for only 15 cases. These cases were treated with ADT or orchidectomy. In this study, we did not interrogate patient matched non-brain metastases, as these samples were not available. Our study design addressed this deficit by comparing the PCBM with the CRPC500-cohort comprising non-brain metastases from multiple sites. All patients with available clinical data harbored metastases in bone and/or lymph nodes and 6/15 patients also presented liver and/or lung metastases. The extent of tumor burden in the PCBM cohort might help to explain the higher mutational burden seen in the brain metastases. Unfortunately, the CRPC500 data does not include data on tumor burden.

13. Figure 1b, the y-axis scale of the one extreme outlier PCA sample can be broken to help scale the rest of the graph more appropriately.

Response #13

We appreciate the suggestion, but have opted not to break the y axis as this breaks the convention that the visual difference in height between bars is proportionate to the difference in values.

Changes to manuscript #13

None.

Reviewers' Comments:

Reviewer #2:

Remarks to the Author:

1. The concerns regarding the methods in determining the mutation calls still remain. The stringency of mutation filter is extremely low in this study (1% VAF and 3 supporting reads as mentioned in Methods, and 0.1% or 1 supporting read as shown in Table SD5), which may not even filter out sequencing errors efficiently. The methods need to be validated first so that the findings and conclusions can be convincing.
2. The observations on NF1 and RICTOR, predominantly subclonal in case of NF1 and all the RICTOR mutations are VUS and non-recurrent, do not support the strong claim in the abstract. Without recurrence or literature support, functional studies are almost essential to suggest that these random RICTOR mutations are oncogenic and druggable.
3. The authors' claims have been watered down in the body of the text, and to be consistent need to be removed from the abstract, or at most mentioned as observations of potential interest, not the potentially misleading assertion "highly significant enrichment of mutations in NF1 (25% cases (6/28), $q = 0.049$, 95% CI = 2.38 – 26.52, OR =8.37) and RICTOR (17.9% cases (5/28), $q = 0.01$, 95% CI = 6.74 – 480.15, OR = 43.7) in PCBM compared to non-brain prostate cancer metastases, suggesting possible activation of the druggable pathways RAS/RAF/MEK/ERK and PI3K/AKT/mTOR, respectively."
4. The section on HR aberrations needs more clarity in presentation and writeup- as it has potential implications in therapy. Please consider summarizing the data in a matrix or some such configuration(see attachment)
5. Please review the numbers provided in the abstract to accurately convey the occurrence of HR aberrations in the PCBM cohort. You may include cases with HR signature but no detected HR gene aberrations aligned with the argument that some of these cases may harbor methylation/ other unknown HR defects not analyzed here.
6. One key observation in the study that is supported by the data is the much higher tumor mutation burden in the PCBM cohort- including both the primary tumor and the mets, compared to TCGA, and mCRPCs (no brain mets)- if corroborated by additional analyses would suggest that higher TMB in PCA may predict potential brain mets. This is worth mentioning in the abstract.

Reviewer #3:

Remarks to the Author:

.

NCOMMS-20-18280B: Point by Point Response to Reviewers' Comments

Reviewer #2

1. The concerns regarding the methods in determining the mutation calls still remain. The stringency of the mutation filter is extremely low in this study (1% VAF and 3 supporting reads as mentioned in Methods, and 0.1% or 1 supporting read as shown in Table SD5), which may not even filter out sequencing errors efficiently. The methods need to be validated first so that the findings and conclusions can be convincing.

Response #1

We accept that this VAF threshold may have been too low, and have changed our filtering strategy to a more stringent 5% as recommended by the reviewer.

Changes to manuscript #1 All findings in the manuscript reflect the use of the 5% VAF threshold suggested.

2. The observations on NF1 and RICTOR, predominantly subclonal in case of NF1 and all the RICTOR mutations are VUS and non-recurrent, do not support the strong claim in the abstract. Without recurrence or literature support, functional studies are almost essential to suggest that these random RICTOR mutations are oncogenic and druggable.

Response #2

We have expanded our cohort to now include 51 patients, representing 168 samples. 105 metastatic, and 63 primary samples. After this addition, and the use of the more stringent VAF threshold suggested, we no longer observe any enrichment for NF1/RICTOR mutations.

Changes to manuscript #2

We have completely removed all reference to enrichment for mutations in these specific genes, and have focussed on the enrichment for alterations affecting the HR pathway.

3. The authors' claims have been watered down in the body of the text, and to be consistent need to be removed from the abstract, or at most mentioned as observations of potential interest, not the potentially misleading assertion "highly significant enrichment of mutations in NF1 (25% cases (6/28), $q = 0.049$, 95% CI = 2.38 – 26.52, OR = 8.37) and RICTOR (17.9% cases (5/28), $q = 0.01$, 95% CI = 6.74 – 480.15, OR = 43.7) in PCBM compared to non-brain prostate cancer metastases, suggesting possible activation of the druggable pathways RAS/RAF/MEK/ERK and PI3K/AKT/mTOR, respectively."

Response #3

Please see response to comment #2

Changes to manuscript #3

Please see response to comment #2

4. The section on HR aberrations needs more clarity in presentation and writeup- as it has potential implications in therapy. Please consider summarizing the data in a matrix or some such configuration (see attachment)

Response #4

We appreciate this helpful suggestion. As part of the major rewriting of the manuscript we have emphasized the significance of our findings regarding HR alterations. (*Note: We never received the attachment*)

Changes to manuscript #4

We have added figure 2a, which summarizes the HR alterations (coding and copy number) across the entire expanded PCBM cohort. We hope this improves the clarity of presentation of this aspect of the study.

5. Please review the numbers provided in the abstract to accurately convey the occurrence of HR aberrations in the PCBM cohort. You may include cases with HR signature but no detected HR gene aberrations aligned with the argument that some of these cases may harbor methylation/ other unknown HR defects not analyzed here.

Response #5

We appreciate that the reviewer agrees with the importance of this finding, we have mentioned this in the abstract.

Changes to manuscript #5

Abstract now reads:

Improved survival in prostate cancer through more effective therapies has also led to an increase of the diagnosis of metastases to infrequent locations such as the brain. In the largest cohort of prostate cancer brain metastases (PCBM) yet studied, we investigated the repertoire of somatic genetic alterations present in brain metastases from 51 PCBM patients. We highlight the clonal evolution occurring in PCBM and demonstrate enrichment of homologous recombination repair (HRR) alteration and homologous recombination deficiency (HRD) related mutational signature (SBS3), concomitant with increased mutational burden in PCBM compared to non-brain metastases. Strikingly, in all patients examined we observed either somatic mutation or copy number loss affecting at least one HRR gene in all metastatic samples, while 10 patients showed homozygous deletion affecting one of these. These features may have potential implications for response to PARP-inhibitor therapy.

6. One key observation in the study that is supported by the data is the much higher tumor mutation burden in the PCBM cohort- including both the primary tumor and the mets, compared to TCGA, and mCRPCs (no brain mets)- if corroborated by additional analyses would suggest that higher TMB in PCA may predict potential brain mets. This is worth mentioning in the abstract.

Response #6 / Changes to manuscript #6

Along with the substantial restructuring of the manuscript, we have also included a reference to this in the abstract (see previous response).

Reviewer #3

1. The selection of VAF < 5% is not sufficiently justified in the manuscript and that it would have been more appropriate to use a more clonal set of variants (>5% VAFs) to determine if PCBMs (primary or mets) represent a distinct molecular entity.

Response #1

We accept that the use of a 1% VAF threshold may be too low, and have now used a threshold of 5% as suggested by the reviewer.

Changes to manuscript #1

All reported findings consider this more stringent threshold.

2. You would need a similar sequencing depth for the non-PCBM cohorts, TCGA and CRPC500, to that of the PCBM's studied here to compare them.

Response #2

We appreciate that differences in coverage may affect the numbers of mutations called. However, in order to ensure total accuracy in this regard we would need access to the raw data from which the CRPC500 data was produced, in order to run it through our pipeline. This would be an extremely lengthy process.

More importantly, we believe the difference in coverage (mean difference ~150x) is not sufficiently large to warrant this, and argue that this is a 'similar sequencing depth'. We note that large-scale WES studies published in high impact journals compare samples sequenced to a range of depths greater than this (e.g. <https://www.nature.com/articles/s41525-017-0021-8/figures/1> and <https://www.nature.com/articles/ng.3126#MOESM9>). Additionally, Numerous high impact studies based on WES have similarly compared mutation calls produced in-house, to TCGA datasets e.g. Bertucci et al (PMID 31118521), Witkiewicz et al. (PMID: 25855536), Wedge et al. (PMID: 29662167). We believe that the key consideration in this regard is that all samples are sequenced to what is generally considered to be sufficient depth for accurate detection of mutations (> 120X). As this is the case across these studies, we do not believe this warrants the complete reanalysis of these datasets.

Changes to manuscript #2

None.

Reviewers' Comments:

Reviewer #2:

Remarks to the Author:

The revised manuscript by Rodriguez-Calero et al., "Enrichment of homologous recombination repair alteration in prostate cancer brain metastases" describes the genetic analysis of 51 prostate cancer cases with brain metastases (PCBM) using WES and targeted RNA sequencing. Three main findings were highlighted. (1) enrichment of homologous recombination repair (HRR) alteration and homologous recombination deficiency (HRD) related mutational signature in PCBM; (2) increased mutational burden in PCBM compared to non-brain metastases, and (3) occurrence of clonal evolution in PCBM. Considering that brain is a relatively uncommon site of metastasis for prostate cancer (according to authors, seen in only 1.5% of metastatic cases), this study represents a logistical feat in putting together high-quality genomic data from so many cases. The study provides a rare glimpse into what is possibly an extreme end of the spectrum of prostate cancer metastatic spread; the data are described with sufficient clarity and figures are self-explanatory for the most part. However, to this reviewer the data doesn't seem to support that PCBMs are substantively distinct from prostate-mets from other sites. And the proposed enrichment of HRR aberrations in PCBMs is based on a loose interpretation of "aberrations"- not compelling biologically and perhaps fraught given its clinical implications. Overall, the molecular alterations were not well discussed. The comparisons between different cohorts (such as PCBM vs. CRPC400 or PCBM primary vs TCGA Prostate) were over generalized without looking into genetic details. Below are the specific concerns to improve this study:

1. While describing a rare subset of prostate cancer mets, comparing its genomic aberrations/ mutation burden, etc. with primary tumors (from TCGA) seems facile and doesn't serve much purpose. The question is if/ how are PCBMs distinct from mets from other sites. Here, even the "primary tumors" from PCBM cases represent a cohort of primary tumors that all went on to metastasize profusely, so comparing them with primary prostate cancer data in TCGA is not ideal, since statistically only a small proportion of those TCGA PCA cases would have gone on to metastasize. To gain insights on brain specific metastases, it is better to compare with primary tumors from CRPC cohorts.
2. Were TCGA prostate and CRPC400 analyzed by the same pipelines as for PCBM? This was not described in the paper. It should be assured that all cohorts are analyzed using the same methods. Number of mutation calls and thereby mutation burden are greatly affected by different filter settings.
3. Were all PCBM samples sequenced by whole exome platform? In the Methods, both whole exome sequencing and target panel sequencing by Ion Torrent were described. This needs to be clarified.
4. The brain mets of the PCBM cohort are derived from autopsies that typically are flush with mets in different organ sites as well, including bones, lymph nodes, liver etc. These mets from cognate patients would arguably make the most ideal comparators to delineate distinctive molecular features of brain mets. The authors do not specifically mention such analyses, it is puzzling why this direct comparison is not performed.
5. The data matrix of aberrations in HRR (PROFOUND) genes- Figure 2- is dominated by simple copy losses that seems to account for the observation of HRR aberrations in 100% of PCBMs (Figure 2), whereas only 20% cases are shown to carry bi-allelic inactivation of an HRR gene. However, mere copy loss only represents heterozygosity of the wild-type gene not associated with loss of function, unless those genes are shown to display haplo-insufficiency (moot question). The strong claim that 100% of PCBMs harbor aberrations in HRR genes based on copy losses is potentially misleading. It is curious if similar criterion is used to call HRR gene aberrations in CRPC500 cohort, whether similarly high numbers would be seen, or perhaps not?
6. It would be more pertinent if CRPC400 and PCBM data are compared for relative frequency of bi-allelic inactivated HRR genes and determine if a significant enrichment is seen in the latter- this will more convincingly support the conclusion that the burden of HRR aberrations drives brain-mets, or at least that it is associated with more aggressive spread, including to remote and less frequent destination sites such as brain.
7. According to the Supplementary Table ST4, there are ~20 patients that carried mutations in the 15 PROfound genes: BRCA1, BRCA2, ATM, BRIP1, BARD1, CDK12, CHEK1, CHEK2, FANCL, PALB2,

PPP2R2A, RAD51B, RAD51C, RAD51D and RAD54L. Among them, 5 patients harbored potentially pathogenic variants (ATM D1467fs, BRCA1 K339fs, BRCA2 R3005*, BRIP1 L54*, and CDK12 K46fs+Q612*). The rest of the mutations are either variants of uncertain significance or known benign variants (such as BRCA2 T1414M and C1290Y). It was stated that 10 patients showed homozygous deletion affecting one of these 15 genes, although the affected genes were not described. Overall, evidence in supporting the enrichment of HRR deficiency in PCBM is weak. (1) In this study, benign and pathogenic alterations are not well distinguished or interpreted differentially. Functionally, benign variants or VUS shouldn't be counted as HRR deficiency unless supported by further evidence. (2) As shown in Supplementary Figure 3, top ranked mutational signatures in PCBM are SBS1, 5, and 16. SBS3 is only present in a small fraction of patients, which is consistent with the molecular results showing in Table ST4. (3) How many patients have a genuine SBS3 mutational signature? Do patients with deleterious mutations or homozygous deletions in BRCA1/2 and PLAB2 show the highest score of SBS3? Loss of the other 12 PROfound genes is not known to result in the SBS3 signature.

8. Supplementary Table ST4: How was LOH in column AG determined? Many mutations with VAF less than 10% are labeled "LOH True", while many mutations with VAF greater than 60% are labeled "LOH False". This indicates that a remarkable fraction of these calls (both SNV and LOH) are not with high confidence and should be investigated carefully and cleaned up.

9. Incidentally, the paper cited by the authors supporting sensitivity of cancers with HRD signature cancer to PARP inhibitors (Poti, A. et al. Genome Biol 20, 240 (2019)- the experimental data is based on cell lines with biallelic deleterious mutations in HRD genes, not merely presence of HRD signature. To suggest that simply the presence of HRD signature can serve as a marker for potential sensitivity to PARP inhibitor (independent of underlying mutations in HRR/HRD genes) is likely to be highly debatable unless supported by experimental data. Until such evidence, it seems problematic to suggest as such in the Discussion: "This study supports the likelihood that men with PCBM may benefit from PARP inhibitors given the high frequency of HR alterations and the prevalence of the HRD signature."

10. Supplementary Figure 2A, B: please make violin plots to better convey the real distribution of the data and highlight the contribution of the outlier hypermutated samples to the significance of the p values.

11. The statement, "Significantly more SNVs, insertions and deletions, were detected in PCBM compared to the matched primaries (SNVs $q = 6.75 \times 10^{-6}$, insertions $q = 3.06 \times 10^{-5}$, deletions $q = 9.30 \times 10^{-3}$, Wilcoxon test) (Figure 1.b, Supplementary Figure 2.a & b).", is not surprising and has been noted to various degrees in all comparisons or primary versus matched mets tissues. It is critical for the argument of this manuscript to compare brain mets with patient-matched other mets.

12. As to the analyses summarized in Figure 3 and encapsulated in the abstract as "We further compared pathways affected by genetic alterations in patients harboring parenchymal brain metastases with those presenting dural metastases in order better to understand the processes which may determine the specific site of metastasis.", going strictly by the available data, it is not necessarily established that metastases to brain are in some way "specific" to brain. The available data may well fit with the null hypothesis that in a bell curve of different organ sites where prostate mets are found, brain represents a relatively rarer destination, potentially associated with a higher metastatic burden overall. This hypothesis can be presumably tested with the available clinical data from the autopsies- and only when this is ruled out, perhaps a brain specific metastatic signature may be admissible.

13. Four of the PCBM patients (P39, P58 P48 and P55) had a mutational burden of >15 mutations/Mb. Patients P39, P48 and P58 had high representation of the SBS44 that could be explained by mutations in MSH2 or POLD1. P55, however, had no clear molecular mechanisms or a distinct mutational signature that could explain the high mutation burden. The authors mentioned mutations in RNMT (RNA guanine-7 methyltransferase) and POLR3C (RNA polymerase III subunit C) in P55, but these genes are not relevant to DNA damage repair machineries. Can authors explain the possible cause of high mutation burden in P55?

Minor comments:

1. Please revise the prepositions used in the first sentence of the Abstract: "Improved survival in

prostate cancer through more effective therapies has also led to an increase of the diagnosis of metastases to infrequent locations such as the brain"

2. Noun subject missing here: "For 20 matched cases with both primary and metastatic samples, we identified putative driver genetic events resulting from clonal evolution that could be responsible for the metastatic and examined the clonal evolution occurring in cases with multiple intratumoral regions within the primary tumor and PCBM."

Reviewer #3:

Remarks to the Author:

Note: i have not been a reviewer on the original paper, and was previously asked to evaluate if reviewer #2 comments were answered.

The comments of the editors and remark #2 of reviewer #2 have only been answered in the letter (with a figure), not, or not clear where, in the manuscript. Furthermore, the sequence depth has only been answered for the DNA not the RNA. Sequence depth has only been answered for the entire batch, not the individual samples.

I am still not convinced that the authors have made the mutation and copy number calling completely comparable to the external cohorts. 1st they do not give an adequate explanation why the VAF calling % could be changed (their choice remains unclear and seems entirely based on the reviewers suggestions, which is not a sound basis). 2nd it does not become clear from the manuscript text for how many samples there was a matched normal. To filter mutations without a matched normal is challenging, requires many many normals WES performed with the same panel, and generally leads to double the amount of called mutations.

From the Figure 1 or methods it does not become clear to me what is RNA fusions and what DNA SNVs. It would be nice to hear if fusions would be valid if confirmed by DNA.

In Figure 1 i would have liked to see the genome wide copy numbers compared to the external cohorts, to see how comparable they are. Now only gene centered copy numbers of few locations are given, not convincing in terms of comparability.

Multiple writing issues, below i give some examples to illustrate how this impedes legibility throughout:

- many sentences are too long, complicated and contain multiple messages which impedes reading (i.e. last sentences introduction). In the last sentences of the intro sentences run over 5 lines (some even 7) and have grammar mistakes (i.e. ""responsible for the metastatic"?. one consequence in intro; it does not become clear what exactly is the definition of the PCBM cohort (the 20 or the 51 cases?). Later in intro the 20 PCBM cases or no longer called PCBM, but matched cases, is this repetition?)

The first line of the results is exactly the same as in the introduction; The same confusing as what is defined as PCBM cohort).

The discussion starts with all kinds of results of a clinical trial published in the NEJM by the Bono. Very confusing.

Figure 2 reports HRD the Figure 2 legends SBS3.

- Code not available, only upon request.

Note from the authors: to help distinguish the changes done in response to the comments from Reviewer 3 and 2, the text changes on the manuscript are highlighted in green and blue, respectively.

NCOMMS-20-18280C-Z: Point by Point Response to Reviewer #3s' Comments

Reviewer #3 (remarks to the authors)

Note: i have not been a reviewer on the original paper, and was previously asked to evaluate if reviewer #2 comments were answered.

1. The comments of the editors and remark #2 of reviewer #2 have only been answered in the letter (with a figure), not, or not clear where, in the manuscript. Furthermore, the sequencing depth has only been answered for the DNA not the RNA. Sequence depth has only been answered for the entire batch, not the individual samples.

Response #1

We have now added the figures to the manuscript as Supplementary Figure 3 and amended the text to describe how the difference in sequencing depth had little impact on TMB.

In regards to the point raised for the RNA, we used our targeted RNAseq assay for the identification of one specific fusion gene. No comparisons were performed using our targeted RNAseq data against TCGA/SU2C cohorts. Thus, we believe this point is not relevant.

For the question related to the sequencing depth, in the revised version of the manuscript, we have generated Supplementary Table 6 detailing the sequencing statistics for individual samples used in this study.

Changes to manuscript #1

Results page 5: These differences persisted after two analyses examining the effect of different sequencing depths across cohorts, first by applying more stringent alternate alle read thresholds (Supplementary Figure 3.a), and secondly by simulating the effects of down sampling our sequencing data (Supplementary Figure 3.b). We also observed the same enrichments when making these comparisons using only the cases for which we had matched normal tissue (Supplementary Figure 3.c). This confirmed that the differences in detected mutations are not as a result of different sequencing depths between cohorts, or the inclusion of samples without a matched normal.

Methods page 19: We carried out two analyses of the effect of differences in sequencing depth between cohorts used in this study, by adjusting the thresholds for alternate allele reads, and by simulating down sampling our sequencing data. The minimum of 1 supporting read in TCGA (sequenced to ~100X) is equivalent to ~2.6 reads at our sequencing depth. Similarly, the minimum of 10 supporting reads in CRPC500 (sequenced to ~160X) is equivalent to ~16 reads at our sequencing depth. We therefore carried out an analysis of our variants using the most stringent combination of thresholds used across these cohorts (i.e. VAF 5% and supported by at least 16 reads) and compared the observed TMB between studies.

To examine the effect of overall sequencing depth on variant calling, we performed a second analysis to simulate the effect of down sampling of our sequencing data to depths (~258X) comparable to TCGA and CRPC500 (the lower being ~100X). Given our current VAF (5%) and supporting read (>3) thresholds, these thresholds would have to be multiplied by 2.6 (~258X/~100X) to simulate the effect of down sampling our ~258X PCBM data to ~100X. This resulted in new thresholds of 12.9% VAF, and a depth threshold of 8 alternate allele reads which were used to filter called variants before repeating the comparison of TMB between studies.

Added Supplementary Figure 3.

2. I am still not convinced that the authors have made the mutation and copy number calling completely comparable to the external cohorts.

1st they do not give an adequate explanation why the VAF calling % could be changed (their choice remains unclear and seems entirely based on the reviewers suggestions, which is not a sounds basis).

Response #2

The change of VAF threshold was based on the reviewer 2's comment that 1% VAF was too lenient for FFPE samples. The switch to the 5% was based on that reviewer's suggestion but also supported by literature. Specifically, Mathieson and Thomas (<https://pubmed.ncbi.nlm.nih.gov/32697619/>) concluded that, on the basis of Prentice et al (<https://pubmed.ncbi.nlm.nih.gov/29698444/>), that FFPE-induced substitutions have VAF<5%. We will also add other papers that support our choice in the revised manuscript. It is important to note that changing the threshold from 1% to 5% did not alter the conclusions of the study. Variant calling, by its very nature, involves defining thresholds to separate real from false variants. The precise choice of 5% is no more or less arbitrary than any other kind of threshold but it is a threshold that makes empirical sense based on what is known about FFPE-induced artifacts.

Changes to manuscript #2

None.

3. 2nd it does not become clear from the manuscript text for how many samples there was a matched normal. To filter mutations without a matched normal is challenging, requires many many normals WES performed with the same panel, and generally leads to double the amount of called mutations.

Response #3

We have added this information to the text in a sentence in the discussion describing this as a limitation of the study. We can confirm that a comparison of the TMB using only the subset of samples with matched germline shows that we still observe the elevated TMB in PCBM compared to TCGA/SU2C - this has been included in Supplementary Figure 3.

Changes to manuscript #3

Discussion page 12: It is important to note that, in 24/51 patients, no matched normal tissue was available, so mutations were called against a pooled normal. However, steps were taken to prevent inflating the number of mutations, and results retained significance when considering only cases for which matched normal tissue was available.

Supplementary Figure 3 added (see above).

4. From the Figure 1 or methods it does not become clear to me what is RNA fusions and what DNA SNVs. It would be nice to hear if fusions would be valid if confirmed by DNA.

In Figure 1 i would have liked to see the genome wide copy numbers compared to the external cohorts, to see how comparable they are. Now only gene centered copy numbers of few locations are nogiven, not convincing in terms of comparability.

Response #4

The fusions are marked in a different color. To further help clarify the distinction, we will rename "TMPRSS2-ERG" to "TMPRSS2-ERG fusion" in Figure 1d, and add a gap in the figure to separate fusion and other types. Unfortunately, given the nature of whole-exome sequencing, we are unable to validate the fusions on the DNA level.

We did not perform any CNA comparison between our cohort and TCGA/SU2C, not on a genomewide level nor on a per-gene level. We see this as out of the scope of this current study.

Changes to manuscript #4

Amended Figure 1d:

5. Multiple writing issues, below if give some examples to illustrate how this impedes legibility throughout:
- many sentences are too long, complicated and contain multiple messages which impedes reading (i.e. last sentences introduction). In the last sentences of the intro sentences run over 5 lines (some even 7) and have grammar mistakes (i.e. ""responsible for the metastatic"?. one consequence in intro; it does not become clear what exactly is the definition of the PCBM cohort (the 20 or the 51 cases?). Later in intro the 20 PCBM cases or no longer called PCBM, but matched cases, is this repetition?)

Response #5

We have revised these complex sentences and corrected errors throughout.

Changes to manuscript #5

Introduction page 3: For 20 matched cases with both primary and metastatic samples, we identified putative driver genetic events resulting from clonal evolution that could drive metastasis and examined the clonal evolution occurring in cases with multiple intratumoral regions within the primary tumor and PCBM.

Results page 3: The PCBM cohort of 51 patients (Supplementary Table ST1, Supplementary Figure 1), represents the largest cohort of this type analyzed to date

Other typos/sentences shortened throughout.

6. The first line of the results is exactly the same as in the introduction; The same confusing as what is defined as PCBM cohort).

The discussion starts with all kinds of results of a clinical trial published in the NEJM by the Bono. Very confusing.

Response #6

We have revised to simplify the message we would like to convey.

Changes to manuscript #6

Introduction page 3: As little is known about the genomic landscape of these rare metastases, we conducted a comprehensive multi-regional genomic analysis of a PCBM cohort of 51 patients, with non-synchronous matched primary samples available for 20 patients.

Results page 3: The PCBM cohort of 51 patients (Supplementary Table ST1, Supplementary Figure 1), represents the largest cohort of this type analyzed to date

Discussion page 12: The PROfound phase 3 trial demonstrated prolonged overall survival in the cohort of patients with mCRPC and *BRCA1*, *BRCA2* or *ATM* when treated with the PARP inhibitor olaparib after progression on enzalutamide or abiraterone. However, patients with known brain metastases were excluded from the PROfound trial¹⁴.

7. Figure 2 reports HRD the Figure 2 legends SBS3.

Response #7

We have revised this in the manuscript

Changes to manuscript #7

Figure 2 legend page 24: Comparison of HRD (SBS3) signature scores, estimated by SigMA

8. Code not available, only upon request.

Response #8

Custom code was written for primarily for data visualization using publicly available R packages. All other algorithms used have been previously published and cited in the manuscript, and only common statistical tests were used. No new algorithm was developed for this study. We would be glad to make the code available on request.

Changes to manuscript #8

n/a

NCOMMS-20-18280C-Z: Point by Point Response to Reviewer #2's Comments

Author's comment: *The aim of this paper, the first large scale analysis of prostate cancer brain metastases, is to convey one major point:*

- *Brain metastases from prostate cancer may respond to PARPi and should be included in subsequent trials in this area.*

For this reason, we may agree with some of the comments raised by the reviewer, but many of these cannot be addressed due to unavailability of additional samples from these patients. However, this conclusion remains valid despite the absence of metastases from other sites.

Reviewer #2

The revised manuscript by Rodriguez-Calero et al., "Enrichment of homologous recombination repair alteration in prostate cancer brain metastases" describes the genetic analysis of 51 prostate cancer cases with brain metastases (PCBM) using WES and targeted RNA sequencing. Three main findings were highlighted. (1) enrichment of homologous recombination repair (HRR) alteration and homologous recombination deficiency (HRD) related mutational signature in PCBM; (2) increased mutational burden in PCBM compared to non-brain metastases, and (3) occurrence of clonal evolution in PCBM. Considering that brain is a relatively uncommon site of metastasis for prostate cancer (according to authors, seen in only 1.5% of metastatic cases), this study represents a logistical feat in putting together high-quality genomic data from so many cases. The study provides a rare glimpse into what is possibly an extreme end of the spectrum of prostate cancer metastatic spread; the data are described with sufficient clarity and figures are self-explanatory for the most part. However, to this reviewer the data doesn't seem to support that PCBM are substantively distinct from prostate-mets from other sites. And the proposed enrichment of HRR aberrations in PCBM is based on a loose interpretation of "aberrations"- not compelling biologically and perhaps fraught given its clinical implications. Overall, the molecular alterations were not well discussed. The comparisons between different cohorts (such as PCBM vs. CRPC400 or PCBM primary vs TCGA Prostate) were over generalized without looking into genetic details. Below are the specific concerns to improve this study:

1. While describing a rare subset of prostate cancer mets, comparing its genomic aberrations/ mutation burden, etc. with primary tumors (from TCGA) seems facile and doesn't serve much purpose. The question is if/ how are PCBM distinct from mets from other sites. Here, even the "primary tumors" from PCBM cases represent a cohort of primary tumors that all went on to metastasize profusely, so comparing them with primary prostate cancer data in TCGA is not ideal, since statistically only a small proportion of those TCGA PCA cases would have gone on to metastasize. To gain insights on brain specific metastases, it is better to compare with primary tumors from CRPC cohorts.

Response #1
We have been working a few years funded by the SPORC for Prostate Cancer to collect more primary cases. Unfortunately, there are only 5 cases from the CRPC500 cohort with primary tumors. We do not believe a comparison with only 5 cases will provide reliable results to address this question. We compare our primary tumors with a high-grade subset within the TCGA-cohort (Supplementary figure 5) and the PCBM-primaries show a significant enrichment for the HRD-signature (SBS3).

Changes to manuscript #1: None

2. Were TCGA prostate and CRPC400 analyzed by the same pipelines as for PCBM? This was not described in the paper. It should be assured that all cohorts are analyzed using the same methods. Number of mutation calls and thereby mutation burden are greatly affected by different filter settings.

Response #2
We did not use the same pipeline, however, in order to address concerns relating to the use of different pipelines, we performed a number of analyses to demonstrate the validity of our findings when comparing across studies. Reviewer #3 had asked a similar question, which we have addressed on point #1 in the point-by-point response (see above).

Changes to manuscript #2: None.

See Changes to manuscript #1 of the response to Reviewer 3.

3. Were all PCMB samples sequenced by whole exome platform? In the Methods, both whole exome sequencing and target panel sequencing by Ion Torrent were described. This needs to be clarified.

Response #3

We apologize for the lack of clarity. WES was performed on all samples and these are the results shown on Supplementary Figure 1. The IonTorrent panel was also performed, which included both DNA and RNA, but we only used the output of the RNA analysis in this study (for calling fusions). We removed the mention of the IonTorrent DNA panel in the methods for clarity.

Changes to manuscript #3:

Methods page 17: cDNA and 10 ng of DNA were used for library preparation with the Ion AmpliSeq Library Kit Plus with a prostate specific custom multiplex RNA³² panel and barcode incorporation (Ion Torrent, Thermo Fisher).

4. The brain mets of the PCBM cohort are derived from autopsies that typically are flush with mets in different organ sites as well, including bones, lymph nodes, liver etc. These mets from cognate patients would arguably make the most ideal comparators to delineate distinctive molecular features of brain mets. The authors do not specifically mention such analyses, it is puzzling why this direct comparison is not performed.

Response #4

We agree with the reviewer that metastases from other sites, from the same patients, would provide an interesting comparison for the PCBM. However, due to the multi-centric and international nature of this cohort, it was not possible to obtain these samples. Furthermore, **this cohort is predominantly from non-autopsy samples (only 8 cases out of 51 cases)**. Since our primary focus was the analysis of brain metastases, we only collected and analyzed tissue from this metastatic location, even when most of the patients harbored metastases in other anatomical regions (ST1). While this is unfortunate, we would argue that the central finding of this study – i.e. that HRD is frequent in PCBM and that patients with these rare metastases should therefore be included in trials using PARPi –, holds true without comparison against cognate metastases, or even against metastases from other studies.

Changes to manuscript #4: None.

5. The data matrix of aberrations in HRR (PROFOUND) genes- Figure 2- is dominated by simple copy losses that seems to account for the observation of HRR aberrations in 100% of PCBMs (Figure 2), whereas only 20% cases are shown to carry bi-allelic inactivation of an HRR gene. However, mere copy loss only represents heterozygosity of the wild-type gene not associated with loss of function, unless those genes are shown to display haplo-insufficiency (moot question). The strong claim that 100% of PCBMs harbor aberrations in HRR genes based on copy losses is potentially misleading. It is curious if similar criterion is used to call HRR gene aberrations in CRPC500 cohort, whether similarly high numbers would be seen, or perhaps not?

Response #5

We used the same criteria when comparing across cohorts. We would reiterate the point that our observation regarding the relevance of investigating PARPi in PCBM remains valid regardless of the thresholds used to define HRD. However, according to the reviewer comments on the biallelic inactivation we have added a comment in the results and discussion sections.

Changes to manuscript #5:

Results page 8 Taken together, these data highlight the prevalence of HRD in PCBM with implications for patient stratification and treatment. **Notably, the subset of 11 patients harboring** biallelic inactivation of at least one of the 15 PROfound HRR genes, represent a clear group of PCBM patients who would benefit from PARPi as established by the PROfound trial.

Discussion page 12 We observed an increased level of somatic alterations in the PCBM compared to the CRPC500 cohort of non-brain prostate cancer metastases, along with an increased frequency of PROfound gene alteration, with biallelic inactivation of at least one of these genes detected in 11/51 patients.

6. It would be more pertinent if CRPC400 and PCBM data are compared for relative frequency of bi-allelic inactivated HRR genes and determine if a significant enrichment is seen in the latter- this will more convincingly support the conclusion that the burden of HRR aberrations drives brain-mets, or at least that it is associated with more aggressive spread, including to remote and less frequent destination sites such as brain.

Response #6

As mentioned, we aimed to convey that HRD was prevalent in PCBM and therefore these patients would benefit from inclusion in trials of PARPi. As suggested by this reviewer, we have toned down our claims. We have added a comment in the results and discussion highlighting the subset of patients with bi-allelic loss of HRR genes.

Changes to manuscript #6:

See *Changes to manuscript #5*

7. According to the Supplementary Table ST4, there are ~20 patients that carried mutations in the 15 PROfound genes: BRCA1, BRCA2, ATM, BRIP1, BARD1, CDK12, CHEK1, CHEK2, FANCL, PALB2, PPP2R2A, RAD51B, RAD51C, RAD51D and RAD54L. Among them, 5 patients harbored potentially pathogenic variants (ATM D1467fs, BRCA1 K339fs, BRCA2 R3005*, BRIP1 L54*, and CDK12 K46fs+Q612*). The rest of the mutations are either variants of uncertain significance or known benign variants (such as BRCA2 T1414M and C1290Y). It was stated that 10 patients showed homozygous deletion affecting one of these 15 genes, although the affected genes were not described. Overall, evidence in supporting the enrichment of HRR deficiency in PCBM is weak. (1) In this study, benign and pathogenic alterations are not well distinguished or interpreted differentially. Functionally, benign variants or VUS shouldn't be counted as HRR deficiency unless supported by further evidence. (2) As shown in Supplementary Figure 3, top ranked mutational signatures in PCBM are SBS1, 5, and 16. SBS3 is only present in a small fraction of patients, which is consistent with the molecular results showing in Table ST4. (3) How many patients have a genuine SBS3 mutational signature? Do patients with deleterious mutations or homozygous deletions in BRCA1/2 and PLAB2 show the highest score of SBS3? Loss of the other 12 PROfound genes is not known to result in the SBS3 signature.

Response #7

The eight patients with detectable SBS3 are shown in Figure 2a. Patients P1 and P30 show biallelic loss of *BRCA2* and high SBS3, but patients P53 and P27 show biallelic loss of *BRCA2* without significant contribution of SBS3. This is shown in figures 1 and 2.

Changes to manuscript #7: None

8. Supplementary Table ST4: How was LOH in column AG determined? Many mutations with VAF less than 10% are labeled "LOH True", while many mutations with VAF greater than 60% are labeled "LOH False". This indicates that a remarkable fraction of these calls (both SNV and LOH) are not with high confidence and should be investigated carefully and cleaned up.

Response #8

As described in the methods, the estimation was done based on the minor CN using FACETS, which is a well know methods to define CNV.

Changes to manuscript #8: None.

9. Incidentally, the paper cited by the authors supporting sensitivity of cancers with HRD signature cancer to PARP inhibitors (Poti, A. et al. Genome Biol 20, 240 (2019)- the experimental data is based on cell lines with biallelic deleterious mutations in HRD genes, not merely presence of HRD signature. To suggest that simply the presence of HRD signature can serve as a marker for potential sensitivity to PARP inhibitor (independent of underlying mutations in HRR/HRD genes) is likely to be highly debatable unless supported by experimental

data. Until such evidence, it seems problematic to suggest as such in the Discussion: "This study supports the likelihood that men with PCBM may benefit from PARP inhibitors given the high frequency of HR alterations and the prevalence of the HRD signature."

Response #9

Thank you for pointing this. We agree and have amended the discussion more accurately to reflect the role of the HRD signature in indicating PAPRI sensitivity.

Changes to manuscript #9:

Results page 7: Therefore, we investigated the prevalence of the HR deficiency (HRD) mutational signature (SBS3) within the cohort. SBS3 is enriched in brain metastases from colorectal cancer¹⁵, has been detected in primary and metastatic PCa, independent of *BRCA1/2* alterations¹⁶ and has been suggested to contribute to the accurate prediction of response to PARP-inhibitor treatment in combination with the presence of alterations affecting HRR genes¹⁷

10. Supplementary Figure 2A, B: please make violin plots to better convey the real distribution of the data and highlight the contribution of the outlier hypermutated samples to the significance of the p values.

Response #10

n/a (as per Editor's comment to the authors)

Changes to manuscript #10: None.

11. The statement, "Significantly more SNVs, insertions and deletions, were detected in PCBM compared to the matched primaries (SNVs $q = 6.75 \times 10^{-6}$, insertions $q = 3.06 \times 10^{-5}$, deletions $q = 9.30 \times 10^{-3}$, Wilcoxon test) (Figure 1.b, Supplementary Figure 2.a & b).", is not surprising and has been noted to various degrees in all comparisons or primary versus matched mets tissues. It is critical for the argument of this manuscript to compare brain mets with patient-matched other mets.

Response #11

Please see response to comment #4. We have previously addressed a similar point in a prior revision.

Changes to manuscript #11: None.

12. As to the analyses summarized in Figure 3 and encapsulated in the abstract as "We further compared pathways affected by genetic alterations in patients harboring parenchymal brain metastases with those presenting dural metastases in order better to understand the processes which may determine the specific site of metastasis.", going strictly by the available data, it is not necessarily established that metastases to brain are in some way "specific" to brain. The available data may well fit with the null hypothesis that in a bell curve of different organ sites where prostate mets are found, brain represents a relatively rarer destination, potentially associated with a higher metastatic burden overall. This hypothesis can be presumably tested with the available clinical data from the autopsies- and only when this is ruled out, perhaps a brain specific metastatic signature may be admissible.

Response #12

As outlined in our response to comment #4, we do not have access to metastases from other sites from the patients in our cohort. However, we agree with the reviewer that a 'brain specific' signature has not been defined by this study. Consequently, it is impossible to define signatures which define the specific site of brain metastases. The work described in this section would be better described as an examination of the features of each site of PCBM, rather than an attempt to find features which would generally define metastases in each region. We also emphasized that this was only an exploratory analysis. We have amended the results better to reflect this.

Changes to manuscript #12:

Results page 10: We then performed the same comparison, after dividing the metastatic-clonal genes based on the dural or parenchymal metastatic location as an exploratory analysis to dissect the mechanisms which are most frequently affected by genetic alterations in the different metastatic locations.

This comparison was carried out on a limited number of samples (nine dural, ten parenchymal) in order to explore the possibility of any differences between the two sites.

13. Four of the PCBM patients (P39, P58 P48 and P55) had a mutational burden of >15 mutations/Mb. Patients P39, P48 and P58 had high representation of the SBS44 that could be explained by mutations in MSH2 or POLD1. P55, however, had no clear molecular mechanisms or a distinct mutational signature that could explain the high mutation burden. The authors mentioned mutations in RNMT (RNA guanine-7 methyltransferase) and POLR3C (RNA polymerase III subunit C) in P55, but these genes are not relevant to DNA damage repair machineries. Can authors explain the possible cause of high mutation burden in P55?

Response #13

We appreciate this point and have examined further the results obtained for P55. We have adjusted the manuscript by describing the high TMB in this patient in the results section.

Changes to manuscript #13:

Results page 6: those from patient P55 harbored missense variants in *POL3RC* and *RNMT*, but no alteration affecting an MMR or HRR gene. However, SBS10b was detected in this patient, associated with large numbers of mutations in samples defined as 'hypermutators'¹².

Minor comments:

14. Please revise the prepositions used in the first sentence of the Abstract: "Improved survival in prostate cancer through more effective therapies has also led to an increase of the diagnosis of metastases to infrequent locations such as the brain"

Response #14

Thank you for this note, we have corrected the text.

Changes to manuscript #14:

Abstract: Improved survival rates for prostate cancer through more effective therapies has also led to an increase in the diagnosis of metastases to infrequent locations such as the brain.

15. Noun subject missing here: "For 20 matched cases with both primary and metastatic samples, we identified putative driver genetic events resulting from clonal evolution that could be responsible for the metastatic and examined the clonal evolution occurring in cases with multiple intratumoral regions within the primary tumor and PCBM."

Response #15

Thank you for pointing this out. We have corrected the text (see response #5 to reviewer 3).

Changes to manuscript #15:

Introduction Page 3. For 20 matched cases with both primary and metastatic samples, we identified putative driver genetic events resulting from clonal evolution that could drive metastasis and examined the clonal evolution occurring in cases with multiple intratumoral regions within the primary tumor and PCBM.

Reviewers' Comments:

Reviewer #4:

Remarks to the Author:

Note: I was not asked to review this manuscript initially. My comments incorporate other Reviewer's response. As a general Reviewer comment, the lack of page/line numbers make the review quite challenging.

This is a tour de force and beautifully performed study. For the most part, it is a very clearly written manuscript representing an important aim, namely the genomic classification of clinically relevant brain metastases in prostate cancer patients. Prior Reviewers #2 and #3 have made a number of editorial and clarifying requests, which appear to be made, however there are still serious problems with the way the data is analyzed and presented.

Abstract, Title and Section: "Homologous recombination deficiency is prevalent in PCBM":

The major issue is that the title and abstract are centered around the HRR alterations and reports "Strikingly, in all patients examined we observed either somatic mutation or copy number loss affecting at least one HRR gene in all metastatic samples...". While this and Supplementary Figure 6's data of 100% PCBM's having HRR alterations may (although I have questions) be technically correct depending on how "alteration" is defined; it is misleading to suggest that the alterations found in this study would lead to PARPi sensitivity. This has been raised by Reviewer #2 Point #5 and #7; and the authors have insufficiently addressed the Reviewer. Regarding Point #5, single copy loss for an HRR gene does not likely cause phenotypic HR deficiency and PARPi sensitivity - pointing out that 11 patients have biallelic loss does not address the question raised. Regarding Point #7, I am in complete agreement that ONLY the variants listed by the Reviewer are oncogenic. The response does not address two of the Reviewer's points - ie most of the variants found in this study in HRR genes are not oncogenic and that 10 patients with homozygous deletions are mentioned in the Abstract but the genes never discussed. It appears the 10 homozygous deletion patients have deletions in ATM, BRCA1, BRCA2, CHEK2, FANCA, RAD51, RAD51B, with some patients having more than one deletion. Given the above points, I feel the title and abstract are misleading given that most patients in this cohort do not actually have biallelic loss in the genes (ie BRCA1/2, perhaps PALB2) that robustly predict response to PARPi. Including B/LB/VUS mutations, single copy loss and all 10 patients with homozygous deletions in other genes is not supported by data in the literature. For example, in the Response to Point #7, the Authors state that P53 has biallelic BRCA2 loss - this patient has a benign SNP in BRCA2 and therefore cannot be considered a patient with biallelic loss in an HRR gene.

Paragraph 2 of section: It is unclear what is meant by "average SBS3 score for mutations" - SBS3 is a signature from a tumor, not for a mutation.

Figure 2.a - The gene names are left off, making the Figure impossible to analyze. It appears that samples with a very small red SBS3 component in Fig 1c are marked SBS3 Yes in Fig 2a; but then samples with larger amounts of SBS3 are not always marked Yes - ie Patient 1, 9, 12, 18 and 30 (which seems to have the highest fraction of SNVs due to SBS3, in fact).

Figure 2b - What is "HRD signature score"? It appears to be a fraction from below 0.0 to 1.0 - this therefore seems different than Figure 1c where SBS3 (and other signatures) are shown as fraction of SNVs. The methods describe LST but I do not see it used anywhere in the Main Text or Figures - is "HRD signature score" LSTs? If yes, this should be explained.

The authors have an ability here to address different biomarkers of HR deficiency and the Venn diagram of overlap between SBS3, LST-high and copy number loss and/or somatic mutation in BRCA1/2 and other HR genes. P1 and P30 have concordant SBS3 and biallelic BRCA2 loss; however, all of the other patients with SBS3 in Fig 1c/2a do not have a mechanism of biallelic loss - do these

patients also have LST-high? At the very least, BRCA1 and RAD51C promoter methylation should be checked. It is not in and of itself a problem as perhaps there are non-mutational mechanisms leading to the HR deficiency phenotype - but the promoter methylation should be performed and this point addressed

Section: "Clonal evolution in PCBM"

Paragraph 4: "These networks, therefore, represent a link between the increased HRD signature observed in the metastatic samples, and the observed clonal expansion." There is no experimental evidence provided for this claim and it should be removed from the Results (the hypothesis is explained well in the Discussion which is a more appropriate place.

Discussion: Please comment on the lack of a use of other HRD signatures, including HRD scores (genomic LOH, NTAI, LSTs) and expression based signatures as a limitation. As mentioned above, the methods describe LST but I do not see it used anywhere in the Main Text or Figures.

Other comment:

1) Microsatellite analysis is described in the Methods, but then in the Results the comment is "Additionally, samples from patients P39 and P48 harbored missense and frameshift mutations, respectively, in MSH2 and both showed high microsatellite instability (MSI-H) status using a clinical molecular diagnostic assay¹¹ (Data not shown). Did all 4 patients with SBS44 have MSI-H status? This is clinically relevant given lack of a clear biomarker in prostate cancer for immune checkpoint blockade. While only 4 samples, it is worth a sentence in the Results - ie all 4 with SBS44 had MSI-H (or not).

NCOMMS-20-18280D: Point by Point Response to Reviewer #4's Comments

Reviewer #4 (remarks to the authors)

Note: I was not asked to review this manuscript initially. My comments incorporate other Reviewer's response. As a general Reviewer comment, the lack of page/line numbers make the review quite challenging.

This is a tour de force and beautifully performed study. For the most part, it is a very clearly written manuscript representing an important aim, namely the genomic classification of clinically relevant brain metastases in prostate cancer patients. Prior Reviewers #2 and #3 have made a number of editorial and clarifying requests, which appear to be made, however there are still serious problems with the way the data is analyzed and presented.

Response: We thank the Reviewer for the supportive comment.

1. Abstract, Title and Section: "Homologous recombination deficiency is prevalent in PCBM":

The major issue is that the title and abstract are centered around the HRR alterations and reports "Strikingly, in all patients examined we observed either somatic mutation or copy number loss affecting at least one HRR gene in all metastatic samples...". While this and Supplementary Figure 6's data of 100% PCBM's having HRR alterations may (although I have questions) be technically correct depending on how "alteration" is defined; it is misleading to suggest that the alterations found in this study would lead to PARPi sensitivity. This has been raised by Reviewer #2 Point #5 and #7; and the authors have insufficiently addressed the Reviewer. Regarding Point #5, single copy loss for an HRR gene does not likely cause phenotypic HR deficiency and PARPi sensitivity - pointing out that 11 patients have biallelic loss does not address the question raised. Regarding Point #7, I am in complete agreement that ONLY the variants listed by the Reviewer are oncogenic. The response does not address two of the Reviewer's points - ie most of the variants found in this study in HRR genes are not oncogenic and that 10 patients with homozygous deletions are mentioned in the Abstract but the genes never discussed. It appears the 10 homozygous deletion patients have deletions in ATM, BRCA1, BRCA2, CHEK2, FANCA, RAD51, RAD51B, with some patients having more than one deletion. Given the above points, I feel the title and abstract are misleading given that most patients in this cohort do not actually have biallelic loss in the genes (ie BRCA1/2, perhaps PALB2) that robustly predict response to PARPi. Including B/LB/VUS mutations, single copy loss and all 10 patients with homozygous deletions in other genes is not supported by data in the literature. For example, in the Response to Point #7, the Authors state that P53 has biallelic BRCA2 loss - this patient has a benign SNP in BRCA2 and therefore cannot be considered a patient with biallelic loss in an HRR gene.

Response #1

We agree with the reviewer that the presented HRD-related genetic alterations in the manuscript should be better characterized. Please find below the clarifications related to this first comment.

For reporting about HRD and potential response to PARPi, we took as reference the 15 HRD-related genes (from now referred as PROfound genes) (*BRCA1, BRCA2, ATM, BRIP1, BARD1, CDK12, CHEK1, CHEK2, FANCL, PALB2, PPP2R2A, RAD51B, RAD51C, RAD51D, and RAD54L*) and the definition of alteration used as inclusion criteria in the PROfound clinical trial (*An alteration was regarded as deleterious if it results in protein truncation (which includes nonsense, frameshift, or consensus splice site alterations), or select missense alterations well-known to be deleterious in ClinVar/BIC databases. Furthermore, larger-scale alterations, such as genomic truncating rearrangements or homozygous deletions, were also classified as qualifying*) (PMID: 32343890). Through the PCBM-cohort, patients fulfilling the inclusion criteria from PROfound are included in Table 1 (10 patients, 19.6%) (s. below), which represent a subset of the alterations shown in Figure 2.c. in the submitted manuscript. In Table 2 (s. below) we listed PCBM-patients harboring a biallelic loss (truncating mutation & CNV-loss) in one of the PROfound genes (8 patients, 15.7%). In Table 3 (s. below) we included PCBM-patients fulfilling the criteria for cohort A in PROfound (5 patients, 9.8%). Altogether, we show that a subset of the PCBM-cohort would have qualified for enrollment in the PROfound clinical trial and, therefore, might have profited from PARPi-therapy, as shown in PROfound results.

These clarifications are now included with the corresponding changes in the title, abstract and manuscript.

Table 1:

Patients fulfilling PROfound criteria

Profound criteria:

A patient had a qualifying alteration if any deleterious or suspected deleterious alteration was found in the 15 pre-specified genes with a direct or indirect role in HRR. An alteration was regarded as deleterious if it results in protein truncation (which includes nonsense, frameshift, or consensus splice site alterations), or select missense alterations well-known to be deleterious in ClinVar/BIC databases. Furthermore, larger-scale alterations, such as genomic truncating rearrangements or homozygous deletions, were also classified as qualifying.

Patients with homozygous deletion						
Gen						
57	PPP2R2A					
27	BRCA2					
30	BRCA2					
45	PPP2R2A					
47	PPP2R2A, CHEK2, RAD54L, BRCA1					
12	PPP2R2A					
Patients with truncating mutations						
Gen	Protein_Change	Variant_Classification	Clonal Status	ClinVar	Other sources	
39	BRCA1	p.Lys339fs	Frame_Shift_Del	clonal	National Center for Biotechnology Information. ClinVar; [VCV000266130.2], https://www.ncbi.nlm.nih.gov/clinvar/variation/VCV000266130.2 (accessed Feb. 26, 2022).	
48	BRIP1	p.Leu54*	Nonsense_Mutation	clonal	National Center for Biotechnology Information. ClinVar; [VCV000935760.2], https://www.ncbi.nlm.nih.gov/clinvar/variation/VCV000935760.2 (accessed Feb. 26, 2022).	
1	BRCA2	p.Arg3005*	Nonsense_Mutation	clonal	National Center for Biotechnology Information. ClinVar; [VCV000234445.8], https://www.ncbi.nlm.nih.gov/clinvar/variation/VCV000234445.8 (accessed Feb. 26, 2022).	
8	CDK12	p.Gln612*	Nonsense_Mutation	subclonal	Not reported	PMID: 33804295
	CDK12	p.Lys46fs	Frame_Shift_Del	clonal	Not reported	
Total of patients: 10/51= 19.6%						

Table 2:

Patients with biallelic loss (homozygous deletion or truncating mutation and heterozygous loss)

Patients with homozygous deletion						
Gen						
57	PPP2R2A					
27	BRCA2					
30	BRCA2					
45	PPP2R2A					
47	PPP2R2A, CHEK2, RAD54L, BRCA1					
12	PPP2R2A					
Patients with biallelic loss (truncating mutations & heterozygous loss)						
Gen	Protein_Change	Variant_Classification	Clonal Status	ClinVar	Other sources	
1	BRCA2	p.Arg3005*	Nonsense_Mutation	clonal	National Center for Biotechnology Information. ClinVar; [VCV000234445.8], https://www.ncbi.nlm.nih.gov/clinvar/variation/VCV000234445.8 (accessed Feb. 26, 2022).	
8	CDK12	p.Gln612*	Nonsense_Mutation	subclonal	Not reported	PMID: 33804295
	CDK12	p.Lys46fs	Frame_Shift_Del	clonal	Not reported	
Total of patients: 8/51= 15.7%						

Table 3:

Patients with homozygous deletion or truncating mutation in BRCA1, BRCA2 & ATM (Cohort A in PROfound study)						
Patients with homozygous deletion	Gen					
	27	BRCA2				
	30	BRCA2				
	47	BRCA1				
Patients with truncating mutations	Gen	Protein_Change	Variant_Classification	Clonal Status	ClinVar	Other sources
	39	BRCA1	p.Lys339fs	Frame Shift Del	clonal	National Center for Biotechnology Information. ClinVar; [VCV000266130.2], https://www.ncbi.nlm.nih.gov/clinvar/variation/VCV000266130.2 (accessed Feb. 26, 2022).
	1	BRCA2	p.Arg3005*	Nonsense Mutation	clonal	National Center for Biotechnology Information. ClinVar; [VCV000234445.8], https://www.ncbi.nlm.nih.gov/clinvar/variation/VCV000234445.8 (accessed Feb. 26, 2022).
Total of patients: 5/51= 9.8%						

Changes to manuscript #1

Title: Potential for PARP-inhibitor therapy in patients with prostate cancer brain metastases

Abstract: Improved survival rates for prostate cancer through more effective therapies have also led to an increase in the diagnosis of metastases to infrequent locations such as the brain. In the largest cohort of prostate cancer brain metastases (PCBM) yet studied, we investigated the repertoire of somatic genetic alterations present in brain metastases from 51 PCBM patients. We highlight the clonal evolution occurring in PCBM and demonstrate an increased mutational burden, concomitant with an enrichment of the homologous recombination deficiency (HRD) mutational signature in PCBM compared to non-brain metastases. Focusing on known pathogenic alterations within homologous recombination repair genes, we find 10 patients (19.6%) fulfilling the inclusion criteria used in the PROfound clinical trial, which assessed the efficacy of PARP inhibitors (PARPi) in homologous recombination deficient prostate cancer. Eight (15.7%) patients show biallelic loss of one of the 15 genes included in the trial, while 5 patients (9.8%) harbor pathogenic alterations in BRCA1/2 specifically. The uncovered molecular features of PCBM may have therapeutic implications, suggesting potential response to PARPi in patients with PCBM.

Manuscript:

Results section (page 8): Therefore, we focused on known pathogenic alterations affecting PROfound-genes. We found 10 patients (19.6%) fulfilling the inclusion criteria used in the PROfound clinical trial of PARP inhibitors (PARPi) in prostate cancer (Figure 2.c, red highlighted), with 8 of these (15.7%) showing biallelic loss of one of the 15 genes included in the trial, while 5 patients (9.8%) harbored well documented pathogenic alterations of BRCA1/2 specifically (Figure 2.b, Supplementary Data 2, Supplementary Data 3, Supplementary Table 3).

Taken together, these data highlight the prevalence of HRD in PCBM with implications for patient stratification and treatment. Most notably, the subset of 10 patients meeting the inclusion criteria for the PROfound trial, represent a clear group of PCBM patients who would benefit from PARPi.

Tables: We included a new Supplementary Table 3.

Figures: We included the above-mentioned information on the bottom of the Figure 2.b.

2. Paragraph 2 of section: It is unclear what is meant by "average SBS3 score for mutations" - SBS3 is a signature from a tumor, not for a mutation.

Response #2 We apologize for the lack of clarity. We agree that this analysis is confusing and, in keeping with our decision to streamline this study to focus on clinically relevant HRR alterations, we have removed this panel.

Changes to manuscript #2

Figure 2c was removed.

3. Figure 2.a - The gene names are left off, making the Figure impossible to analyze. It appears that samples with a very small red SBS3 component in Fig 1c are marked SBS3 Yes in Fig 2a; but then samples with larger amounts of SBS3 are not always marked Yes - ie Patient 1, 9, 12, 18 and 30 (which seems to have the highest fraction of SNVs due to SBS3, in fact).

Response #3

We apologize that the gene names were not included, we have ensured these are visible on figure 2a (now 2c).

The detection of mutational signatures in Figure 1c was performed using `deconstructSigs` and `MutationalPatterns`, while the specific detection of SBS3 reported in Figure 2a was performed using the SBS3-specific method `SigMA`. Consequently, there are differences due to the orthogonal methods used.

We have clarified this in the text.

Changes to manuscript #3**Methods page 20**

Sensitive calling of SBS3 (HRD) alone was subsequently performed using `SigMA`¹⁸, which uses a multivariate analysis to detect the presence of SBS3.

4. Figure 2b - What is "HRD signature score"? It appears to be a fraction from below 0.0 to 1.0 - this therefore seems different than Figure 1c where SBS3 (and other signatures) are shown as fraction of SNVs. The methods describe LST but I do not see it used anywhere in the Main Text or Figures - is "HRD signature score" LSTs? If yes, this should be explained.

Response #4

We apologize for the lack of clarity here. This is again due to the use of `SigMA` to detect the HRD signature (SBS3) specifically. `SigMA` analysis results in a score for a given sample, for the HRD signature specifically, rather than the fraction of mutations assigned to SBS3 using the other methods. For clarity, we have now changed references to this signature, as detected by `SigMA` to 'HRD signature', while referring to SBS3 in relation to the other methods of calling mutational signatures.

Changes to manuscript #4

Results page 7: Therefore, we investigated the prevalence of the HRD mutational signature within the cohort, as well as in TCGA and CRPC500 cohorts, using the computational tool Signature Multivariate Analysis (`SigMA`) for the highly sensitive detection of the HRD signature¹⁸. We found significantly greater representation of the HRD signature in PCBM compared to CRPC500 ($q = 0.041$, Wilcoxon test),

Legend Figure 2: a. Comparison of HRD signature scores, estimated by `SigMA` for TCGA ($n = 495$), PCBM primary ($n = 63$), CRPC500 non-brain metastatic ($n = 411$), and PCBM metastatic ($n = 105$) samples (Wilcoxon test).

5. The authors have an ability here to address different biomarkers of HR deficiency and the Venn diagram of overlap between SBS3, LST-high and copy number loss and/or somatic mutation in BRCA1/2 and other HR genes. P1 and P30 have concordant SBS3 and biallelic BRCA2 loss; however, all of the other patients with SBS3 in Fig 1c/2a do not have a mechanism of biallelic loss - do these patients also have LST-high? At the very least, BRCA1 and RAD51C promoter methylation should be checked. It is not in and of itself a problem as perhaps there are non-mutational mechanisms leading to the HR deficiency phenotype - but the promoter methylation should be performed and this point addressed.

Response #5

We appreciate this comment and the proposal of a Venn diagram. However, since the strongest evidence of a potential benefit for PARPi in a subset of the PCBM-cohort is based on the existence of alterations in the PROfound genes (s. Response 1), we decided to focus in CNV and mutations in Figure 2.

Changes to manuscript #5

None.

6. Section: "Clonal evolution in PCBM"

Paragraph 4: "These networks, therefore, represent a link between the increased HRD signature observed in the metastatic samples, and the observed clonal expansion." There is no experimental evidence provided for this claim and it should be removed from the Results (the hypothesis is explained well in the Discussion which is a more appropriate place).

Response #6

We agree that this was not an appropriate statement for the results and have removed it.

Changes to manuscript #6

Statement removed from results section (page 10).

7. Discussion: Please comment on the lack of a use of other HRD signatures, including HRD scores (genomic LOH, NTAI, LSTs) and expression based signatures as a limitation. As mentioned above, the methods describe LST but I do not see it used anywhere in the Main Text or Figures.

Response #7 We appreciate the point that we could have used other methods to detect HRD. However, as the focus of this paper was the point that a subset of PCBM patients may benefit from PARPi, the most relevant analyses are those relating to genetic alterations known to confer sensitivity to PARPi. As this reviewer, and the previous, have pointed out, not all alterations affecting HR genes are relevant when assessing potential response to PARPi. While we included LST in previous iterations of this paper (it was left in the methods by mistake) we have now excluded these. We agree that it makes little sense to include further analyses relating to HRD, which may or may not be relevant in determining PARPi response.

Changes to manuscript #7

Removed mention of LST from methods as this was removed from reporting.

8. Microsatellite analysis is described in the Methods, but then in the Results the comment is "Additionally, samples from patients P39 and P48 harbored missense and frameshift mutations, respectively, in *MSH2* and both showed high microsatellite instability (MSI-H) status using a clinical molecular diagnostic assay¹¹ (Data not shown). Did all 4 patients with SBS44 have MSI-H status? This is clinically relevant given lack of a clear biomarker in prostate cancer for immune checkpoint blockade. While only 4 samples, it is worth a sentence in the Results - ie all 4 with SBS44 had MSI-H (or not).

Response #8

We apologize for the lack of clarity here. Three samples had high levels of SBS44, P39, P58 and P48. Of these, P39 and P48 had mutations in *MSH2*, along with high MSI-H according to the molecular diagnostic assay. We have clarified this in the text.

Changes to manuscript #8

Results page 6: Of the three samples with high representation of SBS44, samples from patients P39 and P48 harbored missense and frameshift mutations, respectively, in *MSH2* and both showed high microsatellite instability (MSI-H) status using a clinical molecular diagnostic assay¹¹ (**Supplementary Figure 4**). Samples from patient P58 had a frameshift variant in *POLD1* and a missense variant in *RFC4*, and those from patient P55 harbored missense variants in *POL3RC* and *RNMT*, but no alteration affecting an MMR or HRR gene.